# Genomic basis of the giga-chromosomes and giga-genome of tree peony *Paeonia ostii*

Junhui Yuan [1,8] ✉, Sanjie Jiang [2,8], Jianbo Jian [2,8], Mingyu Liu[1,3], Zhen Yue[2], Jiabao Xu[2], Juan Li[1,3], Chunyan Xu[2], Lihong Lin[1,4], Yi Jing[2], Xiaoxiao Zhang[1,5], Haixin Chen[2], Linjuan Zhang[1,4], Tao Fu[2], Shuiyan Yu[1], Zhangyan Wu[2], Ying Zhang[1], Chongzhi Wang[2], Xiao Zhang[1,3], Liangbo Huang[2], Hongqi Wang[2], Deyuan Hong [6] ✉, Xiao-Ya Chen [7] ✉ & Yonghong Hu [1] ✉

Tree peony (*Paeonia ostii*) is an economically important ornamental plant native to China. It is also notable for its seed oil, which is abundant in unsaturated fatty acids such as α-linolenic acid (ALA). Here, we report chromosome-level genome assembly (12.28 Gb) of *P. ostii*. In contrast to monocots with giant genomes, tree peony does not appear to have undergone lineage-specific whole-genome duplication. Instead, explosive LTR expansion in the intergenic regions within a short period (~ two million years) may have contributed to the formation of its giga-genome. In addition, expansion of five types of histone encoding genes may have helped maintain the giga-chromosomes. Further, we conduct genome-wide association studies (GWAS) on 448 accessions and show expansion and high expression of several genes in the key nodes of fatty acid biosynthetic pathway, including *SAD*, *FAD2* and *FAD3*, may function in high level of ALAs synthesis in tree peony seeds. Moreover, by comparing with cultivated tree peony (*P. suffruticosa*), we show that ectopic expression of class A gene *AP1* and reduced expression of class C gene *AG* may contribute to the formation of petaloid stamens. Genomic resources reported in this study will be valuable for studying chromosome/ genome evolution and tree peony breeding.

Tree peony (*Paeonia ostii* T. Hong et J.X. Zhang) is a medium-sized perennial shrub in the family Paeoniaceae native to central China[1–4]. Its grand flowers have particular significance in Chinese culture and history. Known as the 'king of flowers', it has been cultivated for approximately 2000 years[5,6] and is still widely cultivated today. It is the parental species of varieties of modern cultivated peony (*P. suffruticosa*)[7]. Compared with other angiosperms, tree peony has a relatively smaller number of chromosomes (2n = 10), larger chromosome capacity (10–15 μm), and larger genome size (more than 12 Gb). Due to the disadvantage caused by its giga-chromosomes[8] and overusing its root bark as an ingredient of traditional Chinese medicine[5,9–11], the natural population of *P. ostii* is close to extinction.

[1]Shanghai Key Laboratory of Plant Functional Genomics and Resources, Shanghai Chenshan Plant Science Research Center, the Chinese Academy of Science, Shanghai Chenshan Botanical Garden, Shanghai 201602, China. [2]BGI Genomics, BGI-Shenzhen, Shenzhen, Guangdong, China. [3]College of Landscape Architecture and Forestry, Qingdao Agriculture University, Qingdao, Shandong, China. [4]College of Life Sciences, Shanghai Normal University Shanghai, China. [5]College of Landscape Architecture and Arts, Northwest A&F University, Yangling, Shan'xi, China. [6]State Key Laboratory of Systematic and Evolutionary Botany, Institute of Botany, Chinese Academy of Sciences, Beijing 100093, China. [7]State Key Laboratory of Plant Molecular Genetics, CAS Center for Excellence in Molecular Plant Sciences, Shanghai Institute of Plant Physiology and Ecology, University of CAS, Chinese Academy of Sciences, Shanghai, China. [8]These authors contributed equally: Junhui Yuan, Sanjie Jiang, Jianbo Jian. ✉e-mail: yuanjunhuigsly@126.com; hongdy@ibcas.ac.cn; xychen@cemps.ac.cn; huyonghong@csnbgsh.cn

To date, only a fragmented draft genome of *P. suffruticosa* has been reported[12]. Genomic resources for tree peony are limited.

Morphologically, tree peony has a large number of stamens that develop in the form of a centrifugal androecium, a continuous spiral arrangement from leaf to bract, calyx and corolla, and free nuclear mitosis during embryo development, which are similar to some gymnosperms[1,13,14]. Petaloid stamen has been observed in varieties of modern cultivated peony (*P. suffruticosa*) but not in ancestral species of *P. ostii*[2,15]. However, the mechanism underlying petaloid stamen development in cultivated peony is unclear.

The seed oil of *P. ostii* contains a high proportion (more than 90%) of unsaturated fatty acids, such as α-linolenic acid (ALA), which are essential fatty acids that are beneficial to but not produced by humans[16,17]. Studies of *P. ostii* and other plants have shown that the fatty acid biosynthesis pathway is conserved in land plants, but it is unclear why tree peony seeds can accumulate such high levels of linoleic acid and ALA[16].

Here, we report the chromosome-level assembly of *P. ostii*, showing that this species has a genome size over 10 Gb without additional lineage-specific whole-genome duplication (WGD) in angiosperms. In addition, we discuss the possible factors involved in the establishment and maintenance of the giga-chromosomes. Furthermore, we reveal candidate genes associated with unsaturated fatty acids biosynthesis and the possible reasons leading to the formation of petaloid stamens.

## Results

### Genome assembly and annotation of *P. ostii*

To construct a high-quality genome of tree peony, we first collected DNA from *P. ostii* originated from Luoyang, Henan Province, China. The genome size was estimated through flow cytometry as 12.76 Gb (Supplementary Fig. 1a, b). A total of 2.97 Tb of clean Illumina short-read sequencing data (247x coverage), 643.67 Gb of PacBio long-read sequencing data (53.6x coverage), and 2.50 Tb of Hi-C data were generated (Supplementary Tables 1–3). De novo assembly of the *P. ostii* genome was performed with a combination of SOAPdenovo and wtdbg to obtain the consensus assembly for Pilon correction (Supplementary Fig. 1c). The final assembled genome size was 12.28 Gb with a contig N50 of 228 kb and scaffold N50 of 2.43 Mb (Table 1 and Supplementary Tables 4–7).

Approximately 11.49 Gb (93.5%) of the sequences were anchored onto five chromosomes, consistent with the results of cytological

**Table 1 | Summary of the *P. ostii* genome assembly and annotation**

| Item | Value |
| --- | --- |
| Estimated genome size | 12.76 Gb |
| GC content | 32.80% |
| N50 length (contig) | 228 kb |
| Longest contig | 2.24 Mb |
| Total length of contigs | 12.28 Gb |
| N50 length (scaffold) | 2.43 Mb |
| Longest scaffold | 2.56 Gb |
| Total length of scaffolds | 12.33 Gb |
| Transposable elements | 8.4 Gb |
| Predicted genes | 73,177 |
| Average transcript length | 8222.43 bp |
| Average coding sequence length | 794.61 bp |
| Average exon length | 203.68 bp |
| Average intron length | 2001.82 bp |
| Functionally annotated | 59,768 |

analyses, with little overlap between allelic groups (Supplementary Table 8 and Supplementary Fig. 1d). The average GC content of the *P. ostii* genome was 32.8% (Supplementary Fig. 1e). Furthermore, Benchmarking Universal Single-Copy Orthologs (BUSCO) evaluation of genome assembly quality yielded 94.4% orthologous gene set representation (Supplementary Table 9), consistent with the high mapping coverage of the transcriptome (Supplementary Tables 10–12), confirming the relatively high quality of the assembly.

Serial annotation revealed 73,177 protein-coding gene models (including 54,451 high-confidence genes anchored to the chromosome) with an average gene length of 8.22 kb (Supplementary Table 13 and Supplementary Fig. 2a). Functional annotation against public databases showed that 59,768 (84.53%) of the genes matched known proteins (Supplementary Table 14). BUSCO analysis showed complete gene information for 85.5% of the predicted genes (Supplementary Table 15). In addition, a total of 330,511 pseudogenes were identified in the *P. ostii* genome in accordance with its giga-genome. Additionally, the chloroplast and mitochondrial genomes of *P. ostii* were assembled and annotated (Supplementary Figs. 2b, 2c, 3, 4). A total of 15,238 gene families were identified in *P. ostii*, making it one of the species with the largest number of gene families among land plants (Supplementary Fig. 5a–d). Several gene families involved in secondary metabolite biosynthesis, such as isoflavonoid biosynthesis, were enriched in *P. ostii* (Supplementary Table 16 and Supplementary Fig. 5e), and the expanded gene families in this species were mainly related to DNA replication and fatty acid metabolism (Supplementary Table 17 and Supplementary Fig. 5f). A total of 42,131 noncoding RNAs were also annotated (Supplementary Table 18). Finally, we mapped the annotations to the five pseudochromosomes (ranging from 1.78 to 2.56 Gb) of *P. ostii* (Fig. 1).

### Giga-genome evolution

To gain insight into the evolutionary history of *P. ostii*, we compared its genome with publicly available genomes of 12 species (Supplementary Table 19). Phylogenetic analysis of 474 single-copy orthologous genes from these 13 species classified *P. ostii* into Saxifragales (Supplementary Fig. 6a–d), consistent with the results of previous molecular analysis and recent plastome and transcriptome data[18], supporting Saxifragales being closely related to Vitales and a sister group of Rosids. We estimated that Paeoniaceae and Crassulaceae diverged 109 million years ago (102–120.8 Mya), which is in accordance with the previous estimation of 110 Mya based on rbcL analysis[19].

Fourfold degenerate third-codon transversion (4DTv) analysis of genes in *P. ostii* syntenic blocks indicated that the evolutionary history of this species included a recent LTR burst peak and the γ whole-genome duplication (WGD) event common in eudicots that occurred approximately 130 Mya (Fig. 2a and Supplementary Fig. 6e)[20–22]. Moreover, the analysis revealed that *P. ostii* is unique among angiosperms in having a genome size over 10 Gb without additional lineage-specific WGD (Supplementary Table 20).

Transposable elements (TEs) are ubiquitous components of eukaryotic genomes. Notably, 83.51% of *P. ostii* genes with flanking overlap with TEs had Ks <0.2, indicating that TEs might play a critical role in the formation of giga-chromosomes and giga-genome (Fig. 2b and Supplementary Tables 21, 22). LTRs were the most abundant TEs (43.75%), followed by DNA and long interspersed elements (LINEs). Gypsy was the most abundant LTR superfamily (Supplementary Table 23). Meanwhile, the 16 selected plant species showed great diversity at the TE subfamily level and Del in the Gypsy subfamily exhibited significant expansion in *P. ostii*, reaching 70.58% of full-length LTRs (Fig. 2c and Supplementary Data 1). It indicates that variable quantities and subtypes of Gypsy and Copia contribute differently to various genomes. The giga-genomes of wheat and gingko also experienced explosive LTR expansion[23,24]. The LTR burst in *P. ostii* occurred 1–2 Mya, with a peak at 1.4 Mya (Fig. 2b, d). This peak time

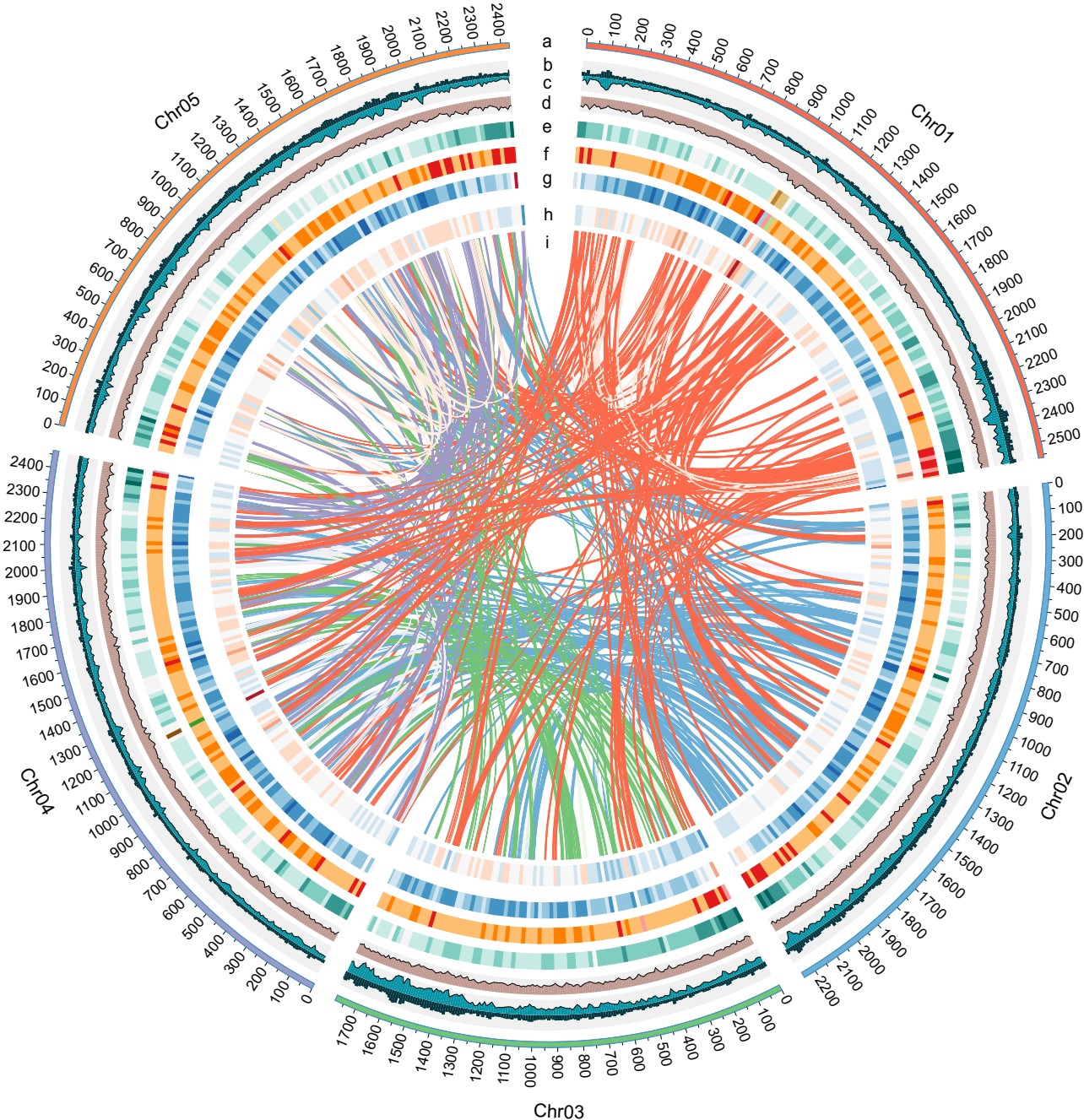

**Fig. 1 | Circos plot of *P. ostii* genomic features. a** Circular representation of the pseudomolecules. **b–d** The distribution of CCGG methylation density, CCWGG methylation density, H3K27me3 enrichment peak density, with densities calculated in 10 Mb windows. **e–h** The distribution of Copia density, Gypsy density, Del density and gene density, with densities calculated in 20 Mb windows. **i** Syntenic blocks. Lines represent links between synteny-selected paralogues.

was in accordance with the insertion time frames of Gypsy and Copia (Supplementary Table 24). Further analysis indicated that the prevalence of five enzymes that are critical for Del generation was 8.25 times higher than that of enzymes critical for the synthesis of other LTRs, which might help explain the explosive Del expansion in *P. ostii* (Fig. 2e). Moreover, the intergenic sequence of the *P. ostii* genome is 15 times larger than that in the genomes of other species with smaller genomes, such as grape and *Arabidopsis*, whereas the lengths of genic regions (such as mRNA, coding sequences (CDSs), and exons/introns) did not differ significantly among these species (Fig. 2f and Supplementary Fig. 6f). Therefore, the replication of intergenic regions due to the insertion of massive LTRs in *P. ostii* was the main driving force for its expanded chromosome size and genome size. The distribution of

TEs and TE subclasses (Gypsy and Del) in *P. ostii* indicated that TEs were mainly inserted into intergenic regions (Fig. 2g), which further confirmed that the burst of LTRs was the critical factor driving its large chromosome and genome sizes. Whole-genome MethylRAD sequencing further indicated that expressed genes, non-expressed genes and pseudogenes showed distinct methylation patterns in the genic and intergenic regions of tree peony (Fig. 2h). The expressed genes were enriched in CCGG methylation but not CCWGG, suggesting that methylation affects the expression types of these genes.

To explore protein–DNA interactions, ChIP-seq was performed, which showed that H3K27me3, the corresponding interacting protein, was enriched in the promoters of paralogues in the TE region compared with the gene region and was more abundant in the promoters

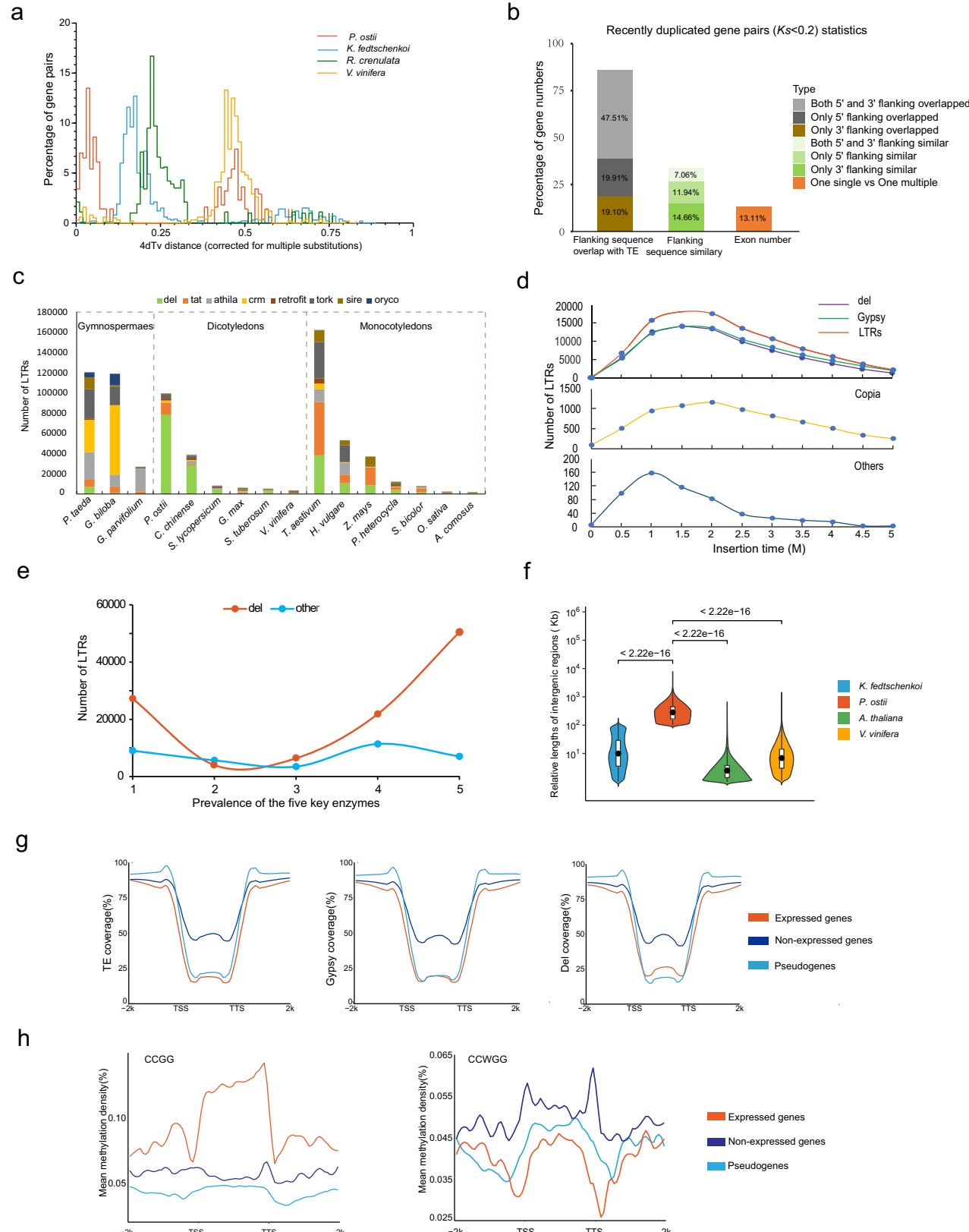

**Fig. 2 | Comparative genomic analysis of *P. ostii*. a** The distribution of 4DTv values, revealing a recent explosion of LTRs and a common γ WGD event in the *P. ostii* genome. **b** Percentage of genes with Ks <0.2 among genes with different characteristics. **c** Subgroups of full-length LTRs in selected Gymnospermae, dicotyledon and monocotyledon species. **d** Estimated insertion times of LTRs and selected subgroups in *P. ostii*. **e** Prevalence of the five key enzymes for LTRs. **f** Relative lengths of intergenic regions in *P. ostii, A. thaliana, K. fedtschenkoi* and *V. vinifera*. Data were presented as the mean values ± SEMs. **g** The distribution of TE coverage, Gypsy coverage and Del coverage in the gene region and the 2k bp external region. **h** Mean methylation density of CCGG and CCWGG of the expressed genes, non-expressed genes and pseudogenes in the peony intergenic regions and the 2k bp external regions. Source data are provided as a Source Data file.

of paralogues in the non-expressed gene region than in the expressed gene region (Supplementary Fig. 6g). These findings suggested that the structure and function of most functional genes were not affected by the giga-chromosomes and giga-genome.

## Giga-chromosome formation

To assess the palaeohistory of *P. ostii*, a comparative genomic investigation of tree peony was performed with *Prunus persica*, *Arabidopsis thaliana*, and *Vitis vinifera* (grape). Seven ancestral eudicot chromosomes containing 7343 protogenes were constructed, and 171 syntenic blocks between tree peony and ancestors were detected (Fig. 3a). Based on the reconstructed ancestral karyotypes (ARK, $n = 7$ and $n = 21$), a minimum of four chromosomal fissions (Cfis) and 20 chromosomal fusions (Cfus) were necessary for peony to reach its current five chromosomes. The extant tree peony was derived from its ancestors without additional polyploidization, whereas *Arabidopsis* (5 chromosomes = 21ARK + 85Cfis − 101Cfus) underwent a much more complex series of duplication events during its evolutionary differentiation from the Brassicaceae ancestor (Fig. 3a).

Chromosomes are mainly formed by DNA and histones. One observation that we speculate might be closely related to the maintenance of the tree peony giga-chromosomes is that the gene families encoding the five types of histones (H1, H2A, H2B, H3 and H4, the major components for chromatin stability) were expanded in *P. ostii* compared with other plant species. Similarly, the expansion of H2A was observed in *Zea mays*, which also has large chromosomes (Fig. 3b, Supplementary Fig. 7a–e and Supplementary Data 2). These 208 histones were distributed among the five giga-chromosomes of *P. ostii* (Supplementary Fig. 7f and Supplementary Data 2). Notably, the genes encoding H2A.W, which promotes chromatin condensation, and H3.1, which is key for chromosome assembly and DNA replication, showed expansion and high expression compared with those of other subgroup proteins (Fig. 3c). The unique variant G at the mutational site 90 of H3.1, which was proven to be related to chromosome assembly and gene silencing, could be found in *P. ostii* and four other plant species (Supplementary Fig. 7g)[25]. Interestingly, 36 out of 38 H3.1 paralogues shared overlapping regions with TEs (Supplementary Table 25).

## GWAS analysis of fatty acid traits

To explore tree peony polyunsaturated fatty acids (PUFA) enrichment at a large populational scale, we resequenced 448 accessions of *P. ostii* from various regions of China (Supplementary Fig. 8a). Approximately 1.34 Tb of raw data were generated by specific-locus amplified fragment sequencing (SLAF-seq), with an approximately 12× tag depth per sample (Supplementary Data 3).

Next, GWASs were conducted to analyse 14 traits related to fatty acid biosynthesis, phenotypic characteristics of fatty acid contents, and other features (Supplementary Table 26 and Supplementary Data 4). C18:0/C18:1$^{\triangle9}$ and C18:1$^{\triangle9}$/C18:2$^{\triangle9,12}$ are two key intermediates in the fatty acid biosynthetic pathway, and the most significant associations between the SNP genotype and these two traits were detected in the regions of chromosomes 2 and 3 (Fig. 4a). A cluster of SNPs present in an approximately 20.02-Mb region on chromosome 3 (655,617 k to 675,640 k with a leading SNP at 673,208,760) was significantly associated with C18:0/C18:1$^{\triangle9}$ (Fig. 4b and Supplementary Fig. 8b), in which nine orthologues of *SAD*, an enzyme transforming 18:0-ACP (acyl carrier protein) to 18:1-ACP, were annotated (Fig. 4c, Supplementary Fig. 8c and Supplementary Data 5). A unique copy, *Pos.gene65901*, was identified in a specific linkage disequilibrium block with a 900-kb range and confirmed to be highly expressed by both transcriptome data and RT–qPCR analysis. The low π value of the region suggests it underwent selection during artificial selection (Supplementary Fig. 8d). A candidate *FAD2* gene (*Pos.gene24209*) on chromosome 2, which converts 18:1-PL (phospholipase) to 18:2-PL, was

highly associated with C18:1$^{\triangle9}$/C18:2$^{\triangle9,12}$ (Fig. 4d, e and Supplementary Fig. 8e, f). Other candidate genes are listed in Supplementary Data 6.

## Endoplasmic reticulum-localised ALA biosynthesis

The content of *P. ostii* fatty acids was then evaluated by mass spectrometry analysis. The essential fatty acids, especially ALA, peaked at the endosperm maturation stage (Fig. 5a). Periodic transcriptomic analysis revealed that the differentially expressed genes were enriched in the unsaturated fatty acid biosynthesis pathway (Supplementary Table 27). In this pathway, several genes acting on key nodes, including *SAD* (stearoyl-ACP desaturase, 13 genes), *FAD2* (fatty acid desaturase 2, four genes), *KAS1* (ketoacyl-ACP synthase 1, three genes) and *FAD3* (fatty acid desaturase 3, four genes), showed expansion and high expression (Fig. 5b, Supplementary Fig. 9a–f and Supplementary Data 7), which might help maintain the high ALA content in tree peony seeds.

Angiosperms have common essential fatty acids, including linoleic acid and ALA, which are synthesised in plastids and assembled in the smooth endoplasmic reticulum (ER) during evolution. The key step of ALA biosynthesis from linoleic acid depends on ω-3 FAD (omega-3 fatty acid Δ15 desaturase), which includes both plastid-localised *FAD7/8* (in green tissue such as leaves, does not contribute to ALA in seed oil) and ER-localised *FAD3* in seeds (contribute to ALA in seed oil). However, the evolution of *omega−3 fatty acid Δ15 desaturase* gene family has not been well illustrated. A WGD event affecting the omega-3 fatty acid Δ15 desaturase occurred in angiosperms, with further divergence in gene function and positioning (Fig. 5c). We found that *FAD7/8* and *FAD3* were widely distributed in both dicots and monocots, including all land plants (Fig. 5c and Supplementary Data 8). A previous study found that, compared with eudicots, almost all monocots, including staple cereals, have a lower content of ALAs[26]. In eudicot species such as peonies and flax, the upregulation of ER-type *FAD3* in a specific stage was associated with high contents of PUFAs. Three *FAD3* genes were cloned for gene function validation. Subcellular localisation confirmed their high expression in the ER (Supplementary Fig. 10a). Results of the yeast expression assay supported that *FAD3_4* (*Pos.gene25040*) was the high-expression copy (Supplementary Fig. 10b and Supplementary Table 28). Together, these findings suggest that expansion and high expression of key node genes in the fatty acid biosynthetic pathway and ER-oriented *FAD3* could drive the high-level accumulation of ALA in the tree peony seeds.

## Development of flowers with petaloid stamens

Following origination from the leaf meristems, the floral meristems of ancestral species of *P. ostii* differentiate into carpels, stamens, petals, sepals, bracts and leaves but not petaloid stamens (Supplementary Fig. 11a). However, parts of the stamens in the cultivated peony (*P. suffruticosa*) can turn into petals during differentiation, which occurs in 60% of cultivars at the squaring stage or the colouring stage[1] (Fig. 6a). Phylogenetic analysis of *P. ostii* and *P. suffruticosa* revealed that multiple homologues of several *MADS-box* family genes, such as *AP1* and *AG* genes (Fig. 6b and Supplementary Fig. 11b), could be critical for flower development[27], especially petaloid stamens. Moreover, the expression patterns of flower development-related genes were similar between *P. ostii* and *P. suffruticosa* (Fig. 6b and Supplementary Fig. 11c). Consistent with the classic model, the class A gene *AP1* showed high expression in petals, and the class C *AG* gene had high expression in stamens. In petaloid stamens, the *AP1* gene was expressed, although at a lower level, and *AG* gene expression was mildly reduced compared to the level in stamens (Fig. 6b), suggesting that the petaloid stamens could result from ectopic *AP1* expression in stamens and *AG* downregulation. This underpins the homeotic transformation of one type of floral organ into another form in cultivated *P. suffruticosa*, i.e. from stamen to petaloids, which increases the flower size and enhances the ornamental value of cultivated peony (*P. suffuticosa*). This unique expression pattern further

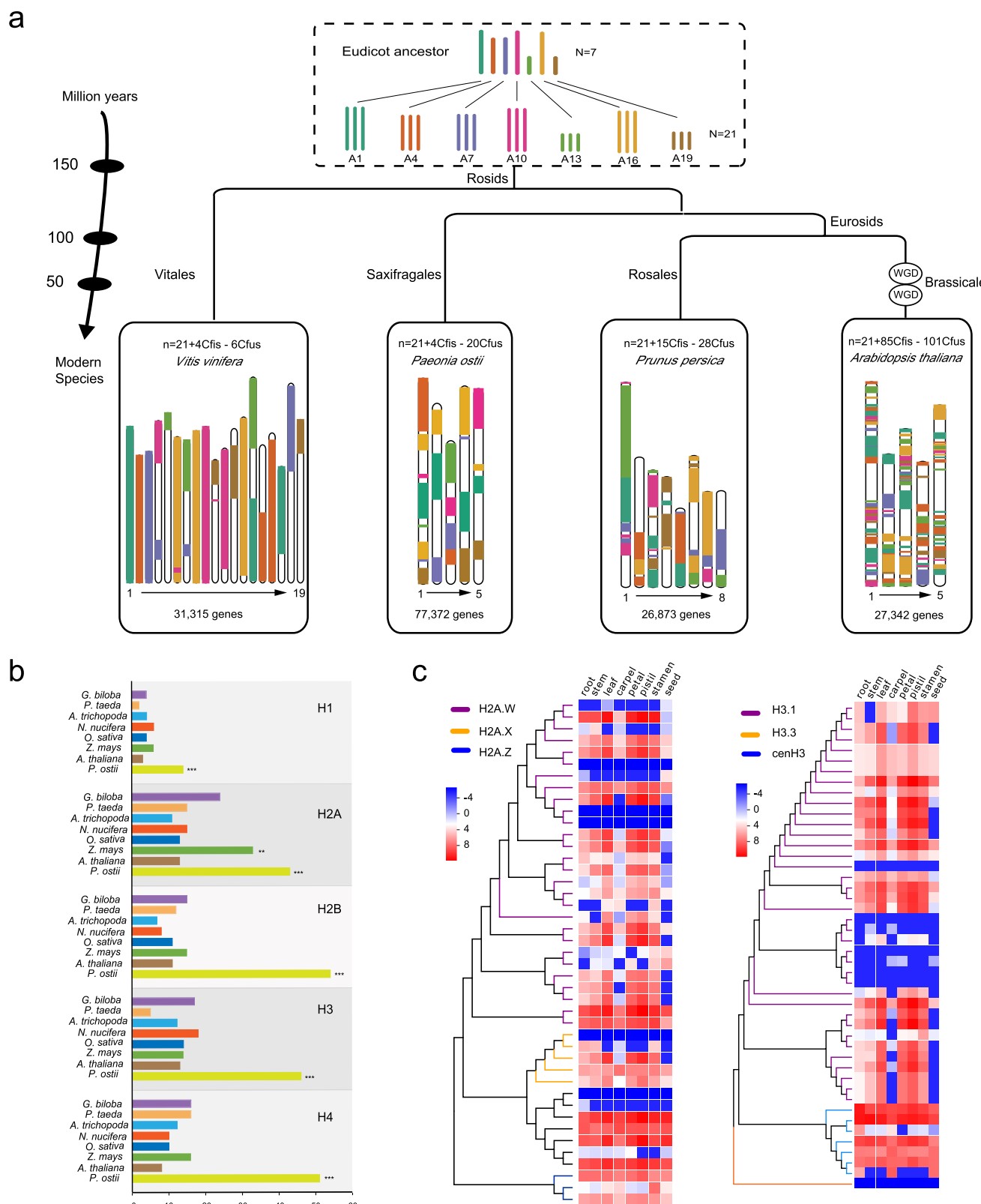

**Fig. 3 | Maintenance of the giga-chromosomes of *P. ostii*. a** Evolutionary scenario of chromosome rearrangements and structural evolution from the eudicot ancestor to tree peony and other plants. The chromosomes are labelled with different colours to indicate the evolution of segments from a common ancestral reconstructed karyotype (ARK, *n* = 7 and *n* = 21) with seven protochromosomes. The chromosome length is zoomed out 100-fold. WGD events are labelled, and the estimated speciation time is indicated on tree branches. **b** Expansion of five types of histones (H1, H2A, H2B, H3 and H4) (**p* < 0.01, ***p* < 0.001). **c** Heatmaps detailing the expansion and expression patterns of histones H2 and H3. The colour of the scale bars represents relative expression. Source data are provided as a Source Data file.

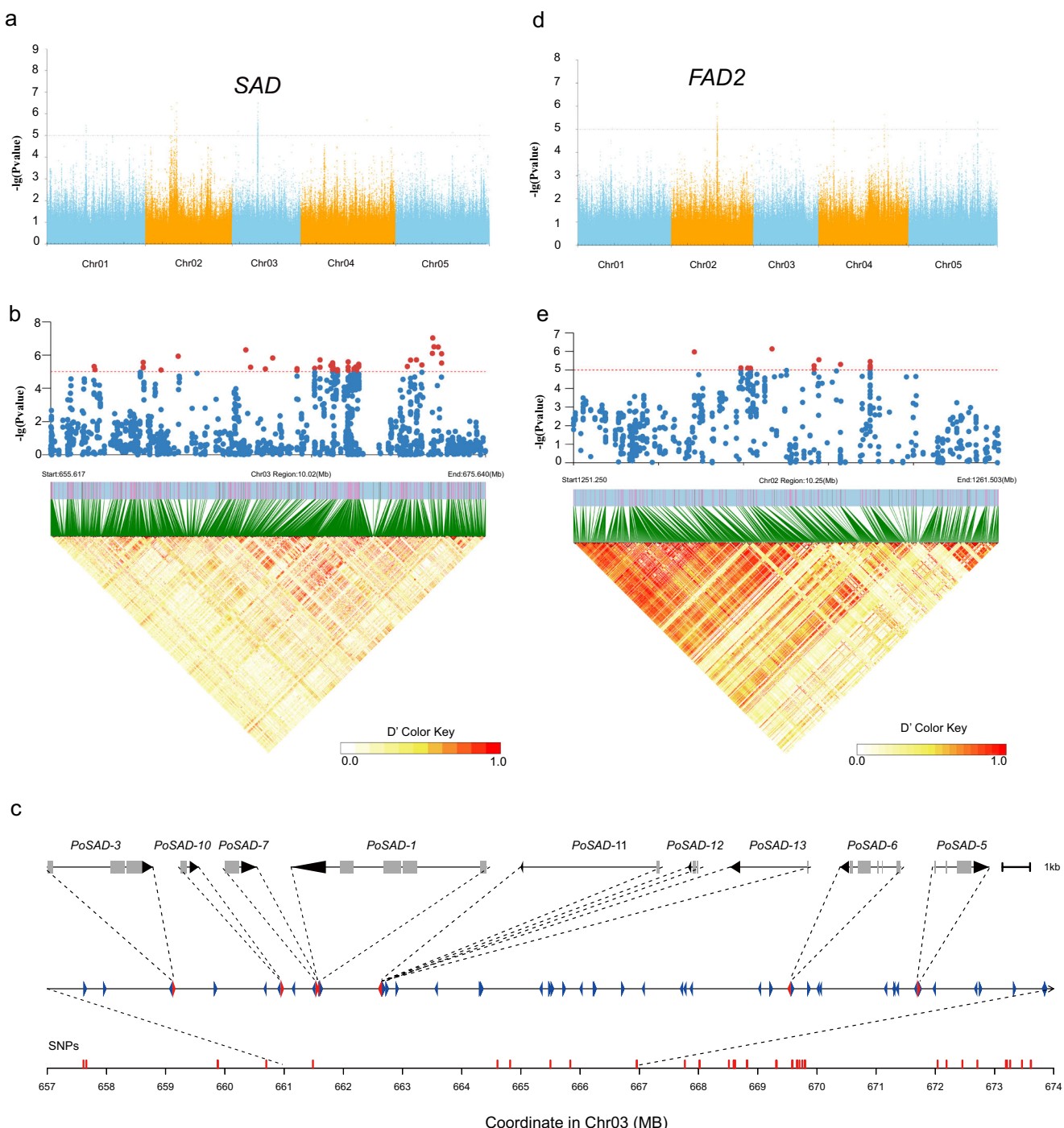

**Fig. 4 | GWAS of fatty acid biosynthesis-related traits in *P. ostii*. a, d** Manhattan plots of genome-wide association mapping results for C18:0/C18:1$^{\triangle 9}$ and C18:1$^{\triangle 9}$/C18:2$^{\triangle 9,12}$. The positions of strong peaks that localised with the known *SAD* and *FAD2* genes investigated in this study are indicated. **b, e** Local Manhattan plot (top) and LD heatmap (bottom) surrounding the peak on chromosomes 3 and 2. **c** A cluster of nine *SAD* genes within a short region of chromosome 3. Source data are provided as a Source Data file.

expands our understanding of the evolutionary direction of the classic ABCE model in core eudicots, such as *Arabidopsis*, which evolved from the 'fading borders' programme of dicots such as *Eschscholzia*. Although *P. ostii* is the ancestral parental species of the derived varieties of *P. suffruticosa*, our results indicated that there might be a 'fading borders' programme in *P. suffruticosa* and the strict ABCE model in *P. ostii*. The rate of ABCE genes' evolution in *P. ostii* and *P. suffruticosa* showed various patterns, while several class A genes experienced

positive selection and class C genes experienced negative selection in *P. suffruticosa* during domestication (Supplementary Fig. 11d). Therefore, the differential flower development pattern between *P. ostii* and *P. suffruticosa*, especially the antagonism of class A genes and class C genes, might have led to petaloid stamens (Fig. 6c). Two genes were examined for their expression in *P. ostii* and *P. suffruticosa* by real-time qRT–PCR to test this model (Fig. 6d and Supplementary Fig. 12a–e). The class A gene *AP1* (*Pos.gene28418*) showed high-level expression in petals,

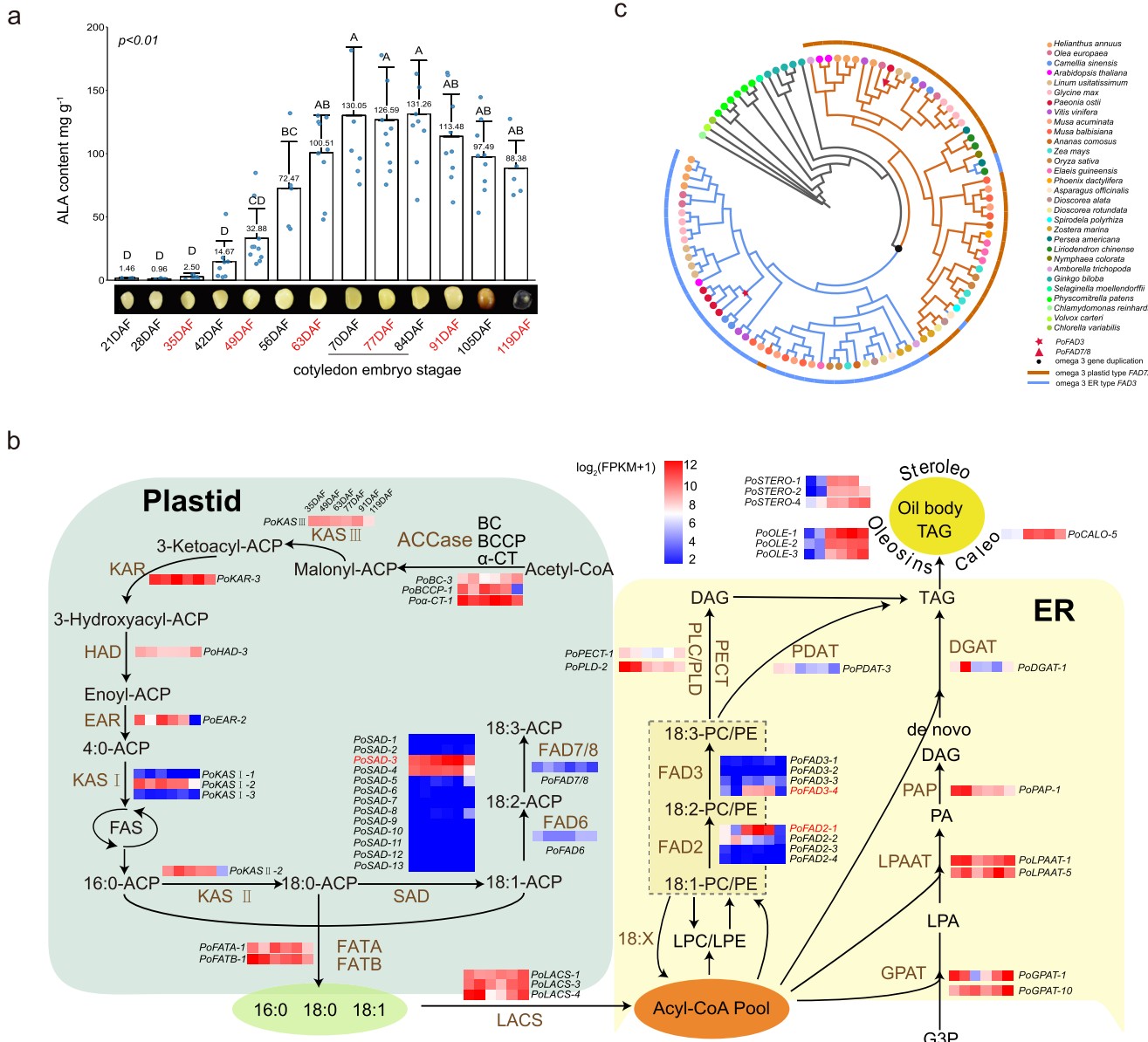

**Fig. 5 | Fatty acid biosynthesis in *P. ostii*. a** Five-year mean ALA contents of peony seeds over the course of development. DAF days after fertilisation. The samples labelled red had transcriptome sequencing data. Data were presented as the mean values ± SEMs. **b** Simplified diagram of the fatty acid biosynthetic pathway with the designated candidate genes. The heatmaps show the candidate gene expression patterns in six stages (35 DAF, 49 DAF, 63 DAF, 77 DAF, 91 DAF and 119 DAF) from the *P. ostii* expression profiles. The major genes are highlighted in red.
**c** Phylogenetic tree of omega-3 desaturases from *P. ostii* and representative plants showing the subfamilies of *FAD3* in the endoplasmic reticulum (ER) and *FAD7/8* in the plastid. Source data are provided as a Source Data file.

low-level expression in petaloid stamens and a negligible level of expression in stamens, whereas the class C gene *AG* (*Pos.gene74744*) exhibited the opposite trend (Fig. 6d) in both *P. ostii* and *P. suffruticosa*. This confirms that the ectopic expression of these genes gave rise to petaloid stamens and suggests a potential breeding trajectory for the development of petaloid stamens phenotypes in peonies.

## Discussion

Many plants are known to have a giga-genome, but it is rare for plants to have a giga-genome composed of giga-chromosomes. The reconstructed ancestral karyotype indicated that a minimum of four chromosomal fissions and 20 chromosomal fusions were necessary for tree peony to obtain its current five giga-chromosomes from the ancestral species without additional lineage-specific polyploidization. We found that the formation of giga-chromosomes of the peony genome seems

to have been driven by large-scale LTR expansion in the intergenic regions. Previous studies on giga-chromosomes have focused on the cytological level, and little is known about the genomic mechanisms of giga-chromosome formation and maintenance. Our results suggested that the *P. ostii* giga-chromosomes (1.78–2.56 Gb) were associated with the expansion of five types of histones. Further experimental evidences, such as functional analysis of histone mutants, are needed to characterise the role of these proteins in maintaining the genome stability of giga-chromosomes.

Plant giga-genomes are commonly formed by two fundamental mechanisms: TE expansion and WGD. In contrast to monocots with giga-genomes, such as wheat[23] and garlic[28], *P. ostii* does not appear to have undergone WGD, making it unique among angiosperms with giga-genomes. Instead, the large chromosome size of the tree peony genome seems to have been driven by large-scale TE expansion in

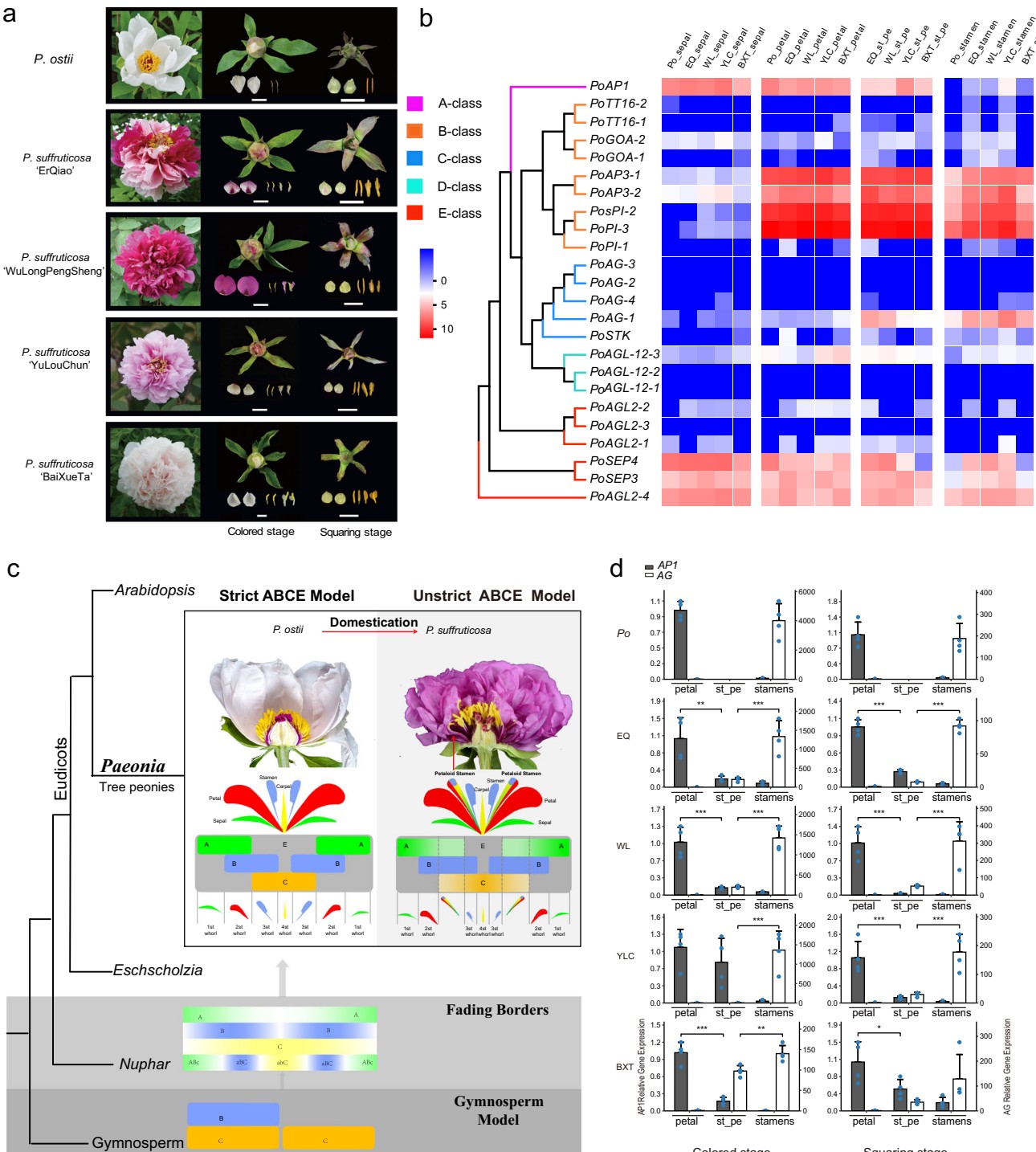

**Fig. 6 | Flower diversity and the development of petaloid stamen in tree peonies. a** Flowers of the tree peony *P. ostii* and four cultivars of *P. suffruticosa* (from top to bottom: 'ErQiao', 'WuLongPengSheng', 'YuLouChun' and 'BaiXueTa') at the blooming, coloured, and squaring stages, respectively (from left to right). Scale bars, 1 cm. **b** Heatmap of the subgroup expansion and expression patterns of flower development-related genes at the coloured stage and squaring stage. Po *P. ostii*, EQ 'ErQiao', WL 'WuLongPengSheng', YLC 'YuLouChun', BXT 'BaiXueTa', st_pe

petaloid stamen. The colour of the scale bars represents relative expression. **c** Schematic diagram of the ABCE model in *P. ostii* and *P. suffruticosa*. **d** Relative expression of the *AP1* gene (class A) and *AG* gene (class C) in petals, petaloid stamens, and stamens measured by real-time quantitative PCR. Four replicates for each gene were performed. Data were presented as the mean values ± SEMs. Source data are provided as a Source Data file.

intergenic regions that favoured the prevalence of five enzymes that are critical for Del generation.

The classic ABCE model depicts the flower differentiation and development of eudicots in several stages, in which there are four morphologically discrete floral organs: sepals are generated when class

A genes act, petals are generated when class A genes and class B genes overlap, stamens are produced by overlapping class B and class C genes, and carpels are produced by class C genes alone[29,30]. We confirmed that the expression of *AG* and *AP1* genes is critical for stamen petalization during flower development in cultivated peony, providing

target genes for the breeding of peony and other ornamental plants with petaloid stamens. The ABCE model suggested that the activity of class A or E proteins alone results in the development of a whorl of green protective sepals and that the activity of B and C proteins together results in the development of the male stamens[15]. In this work, the ectopic expression of *AP1* and *AG* and selective pressure on class A and C genes indicated that there might be a 'fading borders' programme in *P. suffruticosa* and a strict ABCE model in *P. ostii*.

Our genomic and transcriptomic data confirmed the duplication events (*FAD3* and *FAD7/8*) of the omega-3 *FAD* gene family. We not only comprehensively analysed all the genes in the conserved FA synthesis pathway of tree peony but also showed that at least one copy of the genes at each key node in the synthesis pathway was highly expressed. Several genes located on the key nodes, including *SAD*, *FAD2* and *FAD3*, showed expansion and high expression, which might help maintain the high ALA content in tree peony. Artificial modification or insertion of ER-oriented *FADs* into cereals has the potential to increase their production of PUFAs for human health.

## Methods

### Plant materials and genome sequencing

Tree peony (*P. ostii*) originating in Luoyang, Henan Province, was collected from Shanghai Chenshan Botanic Garden and used for de novo sequencing. Various plant tissues, including young leaves and young buds, were dissected for DNA extraction using the DNAsecure Plant Kit (Tiangen). A total of 61 DNA libraries with a gradient of insert sizes were used for Illumina paired-end sequencing, followed by filtering with SOAPnuke (v1.5.5) (https://github.com/BGI-flexlab/SOAPnuke). The 20 kb libraries were constructed for Pacific Biosciences (www.pacb.com) Single-molecule real-time sequencing (SMRT-seq) on both the RSII and Sequel systems, and reads with a length greater than 500 bp were retained for further analysis. Genomic DNA for in situ Hi-C libraries was digested with a suitable four-cutter restriction enzyme (*MboI*), and the ligated fragments were subjected to paired-end sequencing.

### Genome assembly

DNA from fresh leaves of *P. ostii* was used for genome size evaluation by PARTEC CyFlow Space (Germany), and tobacco DNA (~4.5 G) was used as the internal standard. *K-mer* analysis was performed using GenomeScope (v.1.0)[31]. Sequencing reads generated by the Illumina platform were assembled by SOAPdenovo (v2.04)[32] with the parameters '63mer -K 57 -z 20000000000 -R -M 1 -k 31 -F'. The initial PacBio contigs were assembled by wtdbg (v1.2.8)[33] with the following parameters: '-t 50 -i PEO.fa.gz --tidy-reads 5000 -fodbg -k 0 -p 21 -S 4 --rescue-low-cov-edges'. Then, Illumina reads were aligned to the PacBio-corrected contigs with BLASR (v5.1)[34] using '-minMatch13 −maxAnchorsPerPosition 500 −sdpTupleSize 8 −minPctIdentity 75 −bestn 1 −nCandidates 3 −maxScore -500 −noSplitSubreads', SPARC[35] module 'k 2 g 2 c 2 t 0.1 boost 5' and BWA (v0.7.17)[36]. Pilon (v1.23)[37] was further performed to correct errors in the contigs, and SSPACE (v2.1.1)[38] with the parameters '-x 0 -m 32 -o 20 -z 0 -k 5 -a 0.7 -n 15 -v 0' was used to assign the mate-paired reads to scaffolds. The quality of the genome assembly was evaluated by BWA using '-o 1 -e 50 -m 100000 -t 10 -i 15 -I' with random Illumina reads, HISAT2 (v2.0.0-beta)[39] using '-p 8 --phred64 --sensitive --no discordant --no-mixed -I 1 -X 1000' with RNA-seq reads, and BUSCO (v3.0.2)[40] with the final scaffolds. To anchor contigs onto chromosomes, the Hi-C reads were mapped onto the assembled sequences by JUICER (v1.5)[41] and 3D-DNA (v180922)[42] and manually adjusted by JuiceBox (v1.11.08)[43].

### Genome annotation

The repeat contents were determined using a combination of de novo and homology-based approaches. First, LTR_FINDER (v1.0.6)[44], LTRharvest (v1.5.9)[45], LTRdigest (v1.5.9)[46] and RepeatModeler (v1.0.8) (http://www.repeatmasker.org/RepeatModeler/) were used to build a de novo consensus repeat database. Then, RepeatMasker (v4.0.6)[47] was applied to identify repeat elements based on this de novo repeat database. For homology-based identification, RepeatMasker (v4.0.6)[47] and RepeatProteinMask (v.4.0.6)[47] were used to query the Repbase database (v21.01)[48] to identify TEs. Then, the de novo prediction and homologue prediction of repeat elements were combined based on their coordination in the genome. The tandem repeats were annotated using TRF (v4.04)[49].

Gene prediction was based on the integration of homologue evidence-based prediction, de novo prediction, and transcript-assisted prediction. For protein-based homology prediction, protein sequences from closely related or model species, including Arabidopsis (*Arabidopsis thaliana*), apple (*Malus domestica*), *K. fedtschenkoi*, papaya (*Carica papaya*) and grape (*Vitis vinifera*), were aligned to the assembled peony genome using TBLASTN (e-value <1e⁻⁵)[50]. Then, Exonerate (v2.2.0) (--model protein2genome) (https://github.com/nathanweeks/exonerate) was used to map the best hit of alignments and define the gene models. For de novo prediction, the TE regions of the assembled genome were masked using RepeatMasker (v4.0.6)[47]. Then, Augustus (v3.2.1)[51] and SNAP (v2006-07-28)[52] were employed to predict genes using an HMM model trained by predicted peony homologue genes.

Illumina RNA-Seq reads and PacBio Iso-Seq reads were used for protein-based homology searches. Illumina RNA-Seq generated 80 Gb of transcriptome data from eleven peony tissues (apical, carpel, leaf, petal, pistil, seed, stamen, stem, kernel, pericarp, and testa). These reads were mapped to the peony genome using HISAT2 (v2.0.1-beta)[39], and the alignment results were assembled by StringTie (v1.2.1)[53] to obtain the reference-based gene structures. The splice sites were validated and transcripts were assembled again by PASA_lite (v0.1.0)[54] to improve alignment. PacBio Iso-Seq reads were first polished by Quiver (v2.3.2)[55]. The polished consensus sequences were further proofed with Illumina RNA-Seq data using Proovread (v2.14.1)[56]. Then, the corrected full-length transcripts were mapped to the peony genome using GMAP (v2017-11-15)[57] (--min-intronlength=10 --max-intronlength-middle=160000 --max-intronlength-ends=160000 --no-chimeras). The Python script *collapse_isoforms_by_sam.py* from the PacBio repository (https://github.com/PacificBiosciences/cDNA_primer) was used to predict transcript structures and remove redundant transcripts. The splice sites were validated and transcripts were also assembled again by PASA_lite (v0.1.0)[54] to improve alignment. A set of integrated gene models was derived from MAKER (v2.31.8)[58] with Augustus (v3.2.1)[51] and SNAP (v2006-07-28)[52] for gene structure prediction. AED scores were generated for each of the predicted genes as part of the MAKER pipeline, and low-quality genes were filtered by the following conditions: genes with (1) a premature stop codon, (2) a coding sequence (CDS) with ambiguous bases or (3) a CDS length <90 bp.

The stringent confidence-based classification was applied to all predicted genes to discriminate loci representing high-confidence (HC) protein-coding genes and less reliable low-confidence (LC) genes, which potentially consisted of gene fragments, putative pseudogenes or TE-related genes. The genes were classified by the following conditions: (1) genes with significantly high sequence homology (BLASTN e-value <1e⁻¹⁰) to the repeat library and TEs were considered LC genes. (2) Peptide sequences of peony genes were compared against the protein data sets of five homologous species (*Arabidopsis*, apple, *K. fedtschenkoi*, papaya and grape) and the SwissProt database (https://www.uniprot.org/) using BLASTP (e-value <1e⁻¹⁰). The best-matched reference protein was selected as a template sequence. Genes were defined as HC genes if they had a significant BLASTP (e-value <1e⁻¹⁰) hit to reference proteins and over 60% identity to the respective template sequence.

Then, for the LC genes, if they had functional annotations in the KEGG (https://www.kegg.jp/), SwissProt/TrEMBL (https://www.UniProt.org/), GO (http://www.geneontology.org/) or InterPro

(https://www.ebi.ac.uk/interpro/) database with a BLASTP e-value <1e$^{-10}$ and the functions did not match according to 'probable |putative | predicted |uncharacterised |unknown |hypothetical |unnamed', they were transferred to the HC gene set. For genes in the HC gene set, if the function was annotated as 'transposon' or with other TE-related descriptions, then the gene was considered an LC gene. For the genes meeting neither the HC nor LC conditions, if there were supportive functional annotations or RNA-seq/Iso-seq data, they were considered HC genes. Otherwise, they were considered LC genes. The completeness of the genome annotation was assessed using BUSCO (v3.0.2)[40] with the Embryophyta database of 1440 single-copy orthologues.

For gene functional annotation, we aligned the proteins of each gene to the SwissProt/TrEMBL (https://www.uniprot.org/), KOG[59] and NR databases (https://www.ncbi.nlm.nih.gov/refseq/about/nonredundantproteins/) using BLASTP (e-value <1e$^{-10}$), and the function of the best hit was assigned to each gene. We also annotated motifs and domains using InterProScan (v5.11-51.0)[60] by searching against publicly available databases, including Pfam[65], PRINTS[66], PANTHER (http://www.pantherdb.org/), HAMAP[61], Gene3D[62], PROSITE[63], SUPERFAMILY[64], ProDom[65] and SMART[66]. Gene Ontology[67] functional information was retrieved from the NR database by converting NR accession IDs to GO terms. The tree peony genes were also mapped to the KEGG database (Release 84)[68] to find the best hit for each gene. Pseudogenes were identified by the alignment of tree peony protein-coding genes to repeat-masked genomes using Exonerate (v2.2.0) (https://github.com/nathanweeks/exonerate). The predicted hits (with >70% coverage of query proteins) that had frameshifts or premature stop codons compared with the reference proteins were considered pseudogenes. Finally, we identified a total of 330,511 pseudogenes in the tree peony genome; 77,372 were derived from HC genes, and 253,139 were derived from LC genes.

Four types of noncoding RNAs were predicted in the peony genome. The tRNA genes were predicted by tRNAscan-SE-1.23 (v1.3.1)[69] with eukaryote parameters. The rRNAs were identified by aligning the template rRNA (5S, 5.8S and 18S rRNA from *A. thaliana* and 28S rRNA from rice) to the assembled genome using BLASTN with an e-value <1e-5. The miRNA and snRNA genes were predicted using INFERNAL (v1.1.2)[70] by searching against the Rfam database (Release 9.1)[71].

## Giga-genome evolutionary analysis

Gene sets of 12 plant species, namely *A. thaliana* (TAIR10) (https://www.arabidopsis.org/), *Amborella trichopoda*[72], *Aquilegia coerulea*[73], *C. papaya*[74], *V. vinifera*[21], *Solanum lycopersicum*[75], *Nelumbo nucifera*[76], *Jatropha curcas*[77], *Actinidia chinensis*[78], *K. fedtschenkoi*[79], *Helianthus annuus*[80] and *Rhodiola crenulata*[81], were downloaded for evolutionary analysis.

For gene family clustering, BLASTP was used to align the pairwise protein sequences with an e-value <1e$^{-5}$. OrthoMCL (v2.0.9)[82] was used to cluster genes by default parameters with the main inflation value of 1.5. Proteins of single-copy gene families were aligned by MUSCLE (v3.8.31)[83]. Then, fourfold degenerate sites (4dTv) and phase 1 sites of all orthologous single-copy genes were extracted for each species and concatenated into one supergene for phylogenetic reconstruction. The nucleotide sequence of concatenated phase 1 sites was trimmed by trimAl (v1.4)[84] (-gappyout). Then, RAxML[85] was used to reconstruct the phylogenetic tree based on trimmed phase 1 supergene nucleotide sequences with the GTRGAMMA model. To validate the phylogenetic tree, we also constructed phylogenetic trees using PhyML[86] (maximum-likelihood method) based on single-copy genes. The divergence time for the 13 species was estimated based on phase 1 sites of all single-copy orthologous genes. The Markov chain Monte Carlo algorithm for Bayes estimation was adopted to estimate the evolutionary rate and species divergence time using the programme MCMCTree of the PAML package[87]. CAFÉ (v2.1)[88] was used to analyse gene family expansion and contraction. For collinearity analysis, gene sets from

peony[89], *K. fedtschenkoi*[79] and *V. vinifera*[21] were aligned using BLASTP. Then, MCscanX[90] (-k 50 -s 5 -e 1e$^{-05}$ -m 25) was used to construct collinear blocks.

To identify and clarify the WGD events and gene duplications in the peony genome, reciprocal best hit (RBH) paralogues were extracted based on BLASTP alignment (e-value <1e$^{-5}$). The Ks of RBHs was calculated using the Nei−Gojobori method implemented in the yn00 programme of the PAML (v4.8)[87] package. Gene pairs with Ks values less than 0.2 were extracted for TE-related positional and flanking sequence analysis.

We calculated Ka/Ks ratios for the single-copy orthologous genes of the above 13 homologous species. Orthologous genes were first aligned by MUSCLE (v3.8.31)[83]. 'Codeml' in the PAML (v4.8)[87] package was employed with the free-ratio model to estimate Ka, Ks and Ka/Ks ratios on different branches. The difference in mean Ka/Ks ratios for orthologous genes between tree peony and each of the other species was assessed with paired Wilcoxon rank-sum tests. Genes that showed Ka/Ks values higher than 1 along the branch leading to *P. ostii* were reanalysed using the codon-based branch-site model implemented in the PAML (v4.8)[87] package. The pairwise comparisons M1a vs. branch-site model and branch-site model (model = 2, NS sites = 2) vs. branch-site null model (fixed $\omega = 1$ and $\omega = 1$) were used to perform likelihood ratio tests (LRTs). A Bayes empirical Bayes (BEB) analysis was conducted to identify putatively positively selected sites[91]. In addition, significance was evaluated using a $\chi^2$ distribution (df = 1, $p < 0.05$).

## Ancestral chromosome construction and copy numbers of histones

Using the genome of *V. vinifera* as a reference, tree peony homologous gene pairs were identified using BLASTP (e-value <1e$^{-5}$), and the top 10 hits for each gene were retained. Then, the syntenic blocks of the *V. vinifera*[21] genome were screened by MCScanX[90] (http://chibba.pgml.uga.edu/mcscan2/, -e 1e$^{-5}$ -k 50 -g -1 -s 5) based on the identified homologous gene pairs. The seven ancestral chromosomes of eudicots were constructed according to grape genome synteny, and the syntenic blocks between tree peony and the ancestors were detected. Synteny between seven ancestral chromosomes and the chromosomes of other plant species, namely, *Prunus persica*[92], *V. vinifera* and *A. thaliana*, was also analysed.

The copy numbers of histones were investigated in 46 plant species, including two gymnosperms, two basal angiosperms, 10 monocots and 32 dicots. Subgroups of histones from *Arabidopsis*, *Z. mays*[93], *Ginkgo biloba*[94] and peony were aligned by CLUSTALW (v2.1)[95], and the maximum-likelihood phylogenetic tree was built by MEGA (v7.0)[96] with 500 bootstrap replicates.

## Fatty acid measurements and GWAS analysis

In total, 448 individuals of *P. ostii* from five areas, namely, Luoyang, Tongling, Bozhuo, Hezi and Shaoyang of China, were introduced to the special Paeoniaceae nursery at Shanghai Chenshan Botanical Garden (31°4′52″N, 121°10′14″E) in 2014. Over four growing seasons from 2016 to 2019, the budded flowers of each of 448 plants were hand-pollinated with pollen collected from the same *P. ostii* plant, and the seeds were obtained per plant each year. Stored seeds were dried to a constant weight at 60 °C. Dried seeds were pulverised in liquid nitrogen using a mortar and pestle. Approximately 0.2 g of dried powder was weighed and placed in 20 mL screw-cap glass tubes, in which 3 mL of 1:2 chloroform:methanol (v/v) mixture was added. The extraction was performed in a water bath at 35 °C for 1 h at a rotation speed of 120 rpm. After full extraction, 1.0 mL of supplementary chloroform was added to the solution, which was then re-vortexed. Next, 1.8 mL of ddH2O was added so that the final volume ratio of chloroform:methanol:ddH2O was 1:1:0.9. This process was followed by centrifugation at 4000×$g$ for 15 min. The chloroform layer was collected and transferred to another 20 mL screw-cap glass tube. The

entire extraction process was repeated twice. After phase separation, the chloroform layer was withdrawn, dried with sample concentrators under a nitrogen evaporator and stored at -20 °C for further use[16].

The concentrated seed lipids were re-dissolved in 2 mL of $H_2SO_4$ methanol solution (4% $H_2SO_4$)[97]. After charging with nitrogen gas, the sample was vortexed for 1 min and placed in a 90 °C water bath for 1 h. Then, the sample was mixed by vortexing after the addition of 1 mL of $ddH_2O$ and 1 mL of hexane, followed by centrifugation at $4000 \times g$ for 15 min. The supernatant was transferred to a fresh tube, concentrated by bubbling nitrogen and stored at 4 °C for GC–MS analysis. Additionally, 20 µl of nonadecanoic acid (50 mg/mL in hexane) was used as the internal standard for each sample. The FA methyl esters were subjected to GC–MS (GC7890/MS5975, Agilent) on an HP-88 capillary column (60 m × 0.25 mm, 0.2 µm, Agilent). The column temperature was held at 70 °C for 1 min, increased to 210 °C at 10 °C/min for 0 min, 220 °C at 10 °C/min for 0 min and 235 °C at 10 °C/min for 8 min. The injector temperature was set at 250 °C for split injection at a split ratio of 5:1. The injection volume was 1 µl. Helium was used as the carrier gas at a flow rate of 1 mL/min, and the ionisation potential of the mass-selective detector was 70 eV. FA identification was achieved through a mass spectrum database search (NIST MS Search 2.0) and co-elution with a 37-component FAME Mix (Sigma, USA). A standard curve method with an internal standard was used as the quantitative approach to construct three calibration plots of the internal standard peak–area ratio versus standard concentration, as determined by the least squares method. The five major FAs in each sample were quantified in absolute terms using the linear regression of their corresponding standard, while the minor FAs were measured using methyl nonadecanoic acid as the internal standard. FAMEs were expressed as milligrams per gram of DW of the sample. All samples were analysed in triplicate. The statistical analysis method used for data processing was selected according to the procedures[98]. FA contents and percentages were tested by one-way ANOVA ($p < 0.05$), and comparisons between means were performed with Tukey's test.

SLAF-seq libraries were built for a total of 448 young leaf samples for sequencing, and the reads were mapped to the genome sequence by BWA (v0.7.17)[36]. Based on the alignment, we called the SNP variants via the commands 'samtools-1.3.1/samtools mpileup -g -d 100 -q 20 -Q 15 -f ref $bam|bcftools-1.3.1/bcftools call -c -O z - -o $vcf'. According to the genome coordinates, we combined the genotypic traits of all samples with a quality score >20. To identify the high-quality SNPs, SNPs with a minor allele frequency (MAF) <0.05 or missing rate >0.25 were deleted. SHAPEIT (v4.0)[99] was used for imputations and phasing of SNPs. Using the phased SNPs, the genetic distance matrix was calculated according to the p-distance formula, and the NJ tree was constructed via the PHYLIP (v3.696)[100] Neighbour model. Principal component analysis (PCA) of SNPs was performed by GCTA (v1.91.7beta)[101], and ADMIXTURE (v1.3.0)[102] was used for population structure analysis.

To obtain a better distribution for each trait, samples with a trait value outside the range of the scaled mean ± 2 SDs were treated as outliers. To estimate the kinship among samples, TASSEL (v5.0)[103] was used to calculate the kinship matrix. Based on the SNPs, trait values and kinship matrix, we used the FarmCPU model to perform a GWAS with GAPIT (v3.0)[104]. The population structure in the GWAS was estimated in GAPIT with the parameter 'PCA.total=4'. For each trait, we performed a GWAS with the mean trait value and the trait value each year. LDBlockshow (v1.36)[105] was used to identify haplotype blocks based on VCF files with the parameters "-SeleVar 1 -BlockType 3 -MerMinSNP-Num 3 -BlockCut 0.7:0.8". Nucleotide diversity (π) was calculated along the genome in 100-kb nonoverlapping windows by a custom script.

### Transcriptomic analysis of fatty acid biosynthesis pathway genes during seed development

After artificial pollination, seed samples of *P. ostii* were collected weekly (7 days) from seven plants over four growing seasons from 2016 to 2019 (in a total of 13 stages, as shown in Fig. 5a, where the six samples labelled with red were selected for RNA-seq). The seed samples were quickly frozen in liquid nitrogen and stored in a freezer at −80 °C for later use. RNA was then extracted from 12 samples at six stages (35, 49, 63, 77, 91 and 119 DAF) using RNA Exaction Kits (E.Z.N.A. HP Plant RNA Kit, Omega Bio-Tek) and then purified using the RNeasy Plant Mini Kit (Qiagen, Germany) following the manufacturer's protocols. The concentration and quality of each RNA sample were determined using an Agilent 2100 Bioanalyzer (Agilent Technologies, USA). All samples had an OD260/OD280 ratio of 2.0–2.1 and an RNA integrity number >7.0. The extracted total RNA was treated with DNase I, and oligo(dT) primers were used to isolate mRNA. Then, cDNA was synthesised using these mRNA fragments as templates. Short fragments were purified and resolved with EB buffer for end repair. Single A (adenine) nucleotides were then added. Next, the short fragments were ligated with adaptors. Suitable fragments were selected for PCR amplification. The Agilent 2100 Bioanalyzer and the ABI StepOnePlus Real-Time PCR System were used for quality control of the sample library. The generated cDNA libraries were sequenced at BGI-Shenzhen using an Illumina HiSeq 4000 system. HISAT was used for mapping against the reference genome, differential gene expression analysis, and pathway analysis of unsaturated fatty acids in *P. ostii*.

### Evolutionary analysis of fatty acid desaturase-coding genes

Genomic data of 31 representative green plants were downloaded from the NCBI, and key genes for lipid synthesis, including *SAD* (C18:0 to C18:1), Omega6 (*FAD2* and *FAD6*, C18:1 to C18:2) and Omega-3 (*FAD3* and *FAD7/8*, C18:1 to C18:2), were identified by BLAST. A total of 106 amino acids from the FAD genes of these 31 species were aligned with MUSCLE v3.8.31 and then used to create a maximum-likelihood phylogenetic tree in PhyML v3.0. The phylogenetic trees were visualised using the interactive Tree of Life (iTOL).

### qRT–PCR analysis of key genes in ALA biosynthesis

Eight key genes (*FAD3_4*, *FAD7/8*, *SAD_3*, *FAD2_1*, *FAD6*, *CALO_5*, *STERO_4* and *OLE_1*) associated with FA biosynthesis and TAG assembly, particularly those related to ALA biosynthesis, were selected for qRT–PCR validation. The expression levels of these genes were quantified in seeds at 35, 49, 63, 77, 91 and 119 DAF (12 samples in total). qRT–PCR with four replicates for each gene was performed on the ABI StepOnePlus platform with actin as the internal standard. Total RNA was isolated from each tissue at various developmental stages using RNA Exaction Kits (E.Z.N.A. HP Plant RNA Kit, Omega Bio-Tek). First-strand cDNA was prepared from 1 µg of total RNA per sample using a FastKing RT Kit with gDNase (Tiangen). PCRs were performed on an ABI StepOnePlus® Real-Time PCR System (Applied Biosystems) following the manufacturer's instructions. Each reaction mixture (20 µL) contained 10 µL of TB Green Premix Ex Taq II (Tli RNaseH Plus) (Takara), 0.8 µL of each primer (10 µM), 0.3 µL of cDNA template (1 µg), and 8.1 µL of RNase-free water. PCRs for each gene were performed in triplicate, with the following thermal cycling conditions: 95 °C for 30 s, 40 cycles of 95 °C for 5 s and 64 °C for 30 s and 95 °C for 15 s. Primer specificity was confirmed by melting curve analysis. The relative expression levels of the tested genes were calculated using the $2^{-\Delta\Delta Ct}$ method, using the *actin* gene as an internal control.

### Subcellular localisation of FAD3 and functional validation

Fatty acid linoleate desaturases (FAD3 and FAD7/8) are key enzymes that catalyse the production of ALA (C18:3) from LA (C18:2), especially FAD3, in peony seed development. FAD7/8 is located in plastids, and FAD3 is located in the endoplasmic reticulum (ER). The fused expression vector with a green fluorescent protein (GFP) was transferred into *Agrobacterium tumefaciens* using the freeze-thaw method and simultaneously used to infect *Nicotiana benthamiana* leaves.

Based on Gateway technology, expression vectors (pDonr207 and Pyes-DEST52, donated by Prof. Qin Zhao, Shanghai Chenshan Botanical Garden) of FAD3 genes (FAD3_1, FAD3_2 and FAD3_4) were constructed and transferred into *Saccharomyces cerevisiae* INVSc1 (Ura-defects, donated by Prof. Zhi-gang Zhou, Shanghai Ocean University). The transformants (pY31 with FAD3_1, pY33 with FAD3_3 and pY34 with FAD3_4) were selected on SC minimal medium that was deficient in uracil (SC-U) and containing 2% glucose as the sole carbon source. The wild-type and transgenic yeasts were fed LA, and 2% galactose was added as an inducer. The fatty acid profiles of yeast were detected by GC–MS as described above.

## ChIP-seq and DNA methylation analyses

Three replicates of ChIP-Seq were performed to detect genome-wide histone H3 methylation using the anti-histone H3 antibody (tri methyl K27) by Cloud-Seq Biotech (Shanghai, China). Briefly, the yield of ChIPed DNA was determined via a Quant IT fluorescence assay (Life Technologies), and the enrichment efficiencies of ChIP reactions were evaluated by qPCR[106]. Illumina sequencing libraries were generated with the NEBNext® Ultra™ DNA Library Prep Kit (New England Biolabs) by following the manufacturer's manual. The library was checked for quality by an Agilent 2100 Bioanalyzer (Agilent) and then subjected to high-throughput 150 bp paired-end sequencing on an Illumina HiSeq sequencer according to the manufacturer's recommended protocol.

Seeds of *P. ostii* (two biological replicates) were subjected to reduced-sequencing DNA methylation profiling (referred to as MethylRAD) to analyse the genome-wide distribution of methylation sites (CpG and CHG) as well as the position of the methylation sites on various gene functional elements[107]. Each sample was tagged with the type II B enzyme FspEI, and the pooled library was sequenced by Illumina 2000 PE (100–150 bp).

## Flower biodiversity analysis

The flower buds from four traditional tree peony cultivars of *P. suffruticosa* ('ErQiao', 'WuLongPengSheng', 'YuLouChun' and 'BaiXueTa') and its ancestral species *P. ostii* were sampled at the squaring and coloured stages, respectively. Flower buds of similar size at the coloured stage and the squaring stage of the same plant were collected and placed on ice. The petals, petaloid stamens, stamens and dew stage petals were dissected with a microscope (Leica S8APO Leica DFC295). Each category of the samples included four biological replicates. After being treated with liquid nitrogen for 20–30 min, the samples were stored at −80 °C for RNA extraction, sequencing and RT–PCR verification.

An HMMER[86] search was used to run HMM files in order to comprehensively identify MADS-box family genes. PhyML[86] was applied to construct a phylogenetic tree of the two peony ABCDE gene families, and then the branch-site model of PAML[87] software (runmode = 0, fix_omega=0) was used to calculate the ω value of each branch in the tree.

Differential expression of A, B, C, D and E genes related to flower development in four floral tissues (petals, petalized stamens, stamens and dew stage petals) was detected, and the subgroup expansion and expression patterns of flower development-related genes were analysed at the coloured stage and squaring stage. qRT–PCR validation was carried out on 56 tissues of the cultivated peony *P. suffruticosa* and its ancestor *P. ostii* with four replicates.

## Reporting summary

Further information on research design is available in the Nature Portfolio Reporting Summary linked to this article.

## Data availability

The original data were publicly available from the China National Gene Bank under accession number CNP0003098 Genome assembly and annotation were placed in the file [https://ftp.cngb.org/pub/CNSA/data5/CNP0003098/CNS0560369/CNA0050666/]. Source data are provided with this paper.

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

## Acknowledgements

The authors acknowledge funding from the Science and Technology Commission of Shanghai Municipality (14DZ2260400, 14JC1403902 and 21DZ1202000), National Natural Science Foundation of China (31470328) and Special Fund for Scientific Research of Shanghai Landscaping and City Appearance Administrative Bureau (G222415, G192418, G182407, G182406, G172401 G162419, G152424 and G142435). We would like to thank Jiajue Li, Zhongying Kang, Zhenglin Fu and Chunmei Wei for providing experimental materials. We thank Bin Han, Jiankang Zhu, Ji Yang, Hong Ma, Hongzhi Kong, Yuehui He, Qixiang Zhang, Daming Zhang, Heng Zhang, Yuannian Jiao and Linfeng Li for participating in discussions and providing suggestions. We thank all the former and present members of the Genomics and Germplasm Innovation of Tree Peony Group, Lanpeng Ma, Yunpeng Wang, Shaobo Du, Ruimin Nie, Xinying He, Jie Qi, Zujie Yao, Yuping Lv, Zhao Liu, Yu Kong, Binjie Ge and Hongxing Yang, for their help.

## Author contributions

J.Y., D.H., X.-Y.C. and Y.H. conceived and designed the project. Y.H. led the project. J.Y. coordinated the project as the chief executive scientist. J.Y. and S.J. wrote the manuscript with major contributions from Y.H. and X.-Y.C. gave suggestions on revising the manuscript. J.Y., M.L., J.L., L.L., L.Z., X.Z., S.Y. and Y.Z. collected the samples and performed the wet lab experiments. Data analysis was led by J.Y. and J.J. or Z.Y., with major contributions from J.X., M.L., C.X., J.L., Y.J., L.L., T.F., Z.W., H.C., S.J., C.W., H.W. and L.H. Interpretation of data and results was led by J.Y. with major contributions from Y.H. and X.-Y.C.

## Competing interests

The authors declare no competing interests.
