## [Peer Review File · Nature Communications]

Genomic basis and evolutionary adaptation of giga-chromosome and giga-genome of tree peony *Paeonia ostii*Reviewers' Comments:

Reviewer #1:

Remarks to the Author:

This study reported a large genome of *P. ostia* based on huge amount of PacBio and Hi-C data.

1. Although the mapping rate of RNA-seq is a little low, the quality of assembly is overall good since the genome is very large. How about the heterozygosity? I did not see the information. If it is high, the assembly would be challenge and the genome quality will be affected.
2. The analysis on fatty acid synthesis has been done in many species and the pathway and key genes are well -known. Thus, this part provides limited novelty and progress.
3. One major concern is about fatty acid profiling and GWAS analysis on fatty acids. First, for fatty acids in seed oil, there are just a few major types. Authors seemed to search the NIST database and identified a lot of minor ones. I believe some fatty acid species (such as C14:0, C17:1 and so one) is wrong by searching the database (the software tells wrong prediction). I suggest authors just analyze major species: C16:0, C18:0, C18:1, C18:2, C18:3 and so on. Second, authors have not presented the distribution of these fatty acid traits, we can not just the quality of data. Third, author should do GWAS for each fatty acid. It seems that authors have not done this for major ones. Is it because the fatty acid composition or content is not accurate? Then, they may do GWAS for the ratio of fatty acids. Fourth, authors showed SAD and FAD2 in the loci identified by GWAS. How about FAD3 in any loci? Since ALA directly synthesized by FAD3 is the focus of this study. Anyway, this part is very unsatisfactory.
4. The method for fatty acid was not well described. How was the methylation done? Temperature and methylation time? Which fatty acid was used as standard? Internal or not? How about fatty acids analysis for the GWAS population? Authors failed to provide any information about sampling, replicates and so on.
5. Line 89 to 90, fatty acid synthesis is conserved. We know FAD2 and FAD3 are key genes determining the content of linoleic acid and α -linolenic acid (ALA). It is certainly the same in *P. ostia*. I believe it is very likely previous studies have some information on this (such as gene cloning, transcriptome analysis). Authors should check if any works have studied linoleic acid and α -linolenic acid synthesis.
6. Line 269 to 292, it is true ALA can be synthesized in plastid by FAD7/8 (in green tissue such as leaves, do not contribute to ALA in seed oil) and in ER by FAD3 in seed (contribute to ALA in seed oil). Authors may not be quite clear about this and some of the writing is misleading. If they want to address ALA in seed oil, they should focus on FAD3. If they want to talk about ALA in leaves and other green tissues (ALA is essential for membrane lipids and related to many biological functions), then FAD7/8 should be analyzed. In this case, Fig 5 may focus on ALA in seed oil and FAD6, FAD7/8, may not be shown in Fig 5.
7. For results in Supplemental Fig. 8, authors need to calculate the composition or content of produced ALA. Not just showing a few pictures.

Some minor issues:

How many SNPs were used for GWAS? Authors should provide such key information.

The method for yeast experiment was not provided. And so fatty acid analysis for yeast.

Line 717, why did authors describe qRT-PCR here?

Line 75, value. It is also the parental

Line 83, species of *P. ostia*. Genomic information would

Line 85, the information is wrong. It should be "a high proportion (more than 90%) of unsaturated fatty acids. Authors need to understand the difference between unsaturated fatty acids and polyunsaturated fatty acids (PUFAs).

Line 204, and giga-genome.

Line 222, with other plant species. Similarly, the expansion of H2A. Authors need to avoid the using of ";" in the manuscript

Line 267, high ALA content

Line 514, K-mer analysis

Line 706, *P. ostia* were

Reviewer #2:

Remarks to the Author:

In this manuscript titled "Genomic basis and evolutionary adaptation of giga-chromosome and giga-genome of tree peony *Paeonia ostia*", Yuan and colleagues present the genome of tree peony *Paeonia ostii*, with the largest chromosome in all sequenced plants. The authors also identified genes for PUFA biosynthesis by GWAS. Overall, this manuscript is interesting and well-written. The genome assembly and GWAS data could be very helpful for the plant community.

One interesting feature of the *Paeonia ostia* genome is its large chromosome. The author proposed that the expansion of histone variants contribute to the giga chromosome size. While the result is preliminary and could be strengthened by performing a statistic analysis to examine if the association between number of histone variants (Table S26) and chromosome size is significant or not.

For the genome annotation, *Paeonia suffruticosa* (PMID: 32551041) contains 34,854 protein coding genes, which is much less than 73,177 protein coding gene that reported in this study. I suggest the authors investigate this big differences of the number between the two related tree peony species. And how many of these protein coding genes are supported by RNA-seq?

I am also amazed by the extremely large number (330,511) pseudogenes identified in *P. ostii* genome. Could the author do some analysis to find some feature/pattern for these pseudogenes, to explain the largest number of pseudogenes to date.

Minor points

- The circos plot in Figure 1 is common for genome paper, but I think this figure didn't convey much information for the audience. I suggest reducing size of this figure, and not necessary to label so many numbers for genome coordinates.
- The replicates number of sequencing library should be mentioned in the method for all related experiments.
- Methods for ChIP-seq and DNA methylation libraries preparation and subsequent analysis are missing. Most method details for Illumina, PacBio, and Hi-C libraries preparation and sequencing are missing.
- Method for identification of pseudogene is missing.
- Method for subcellular localization is missing.
- The author should describe the 448 samples used for GWAS in details.
- L502, "various plant tissues" should provide the name for each tissue.
- L503 "standard protocols from the manufactures", please add a reference or provide the detail method.
- L514, change "K-meranalysis" to "K-mer analysis". L555, change "PacBioIso-Seq" to "PacBio Iso-Seq". In fact, many spaces between word are missing. The authors should check through the manuscript.
- In Fig 2f, remove "1e6" on the top.
- What is colour bar's meaning for Fig.3c and 6b.
- Fig. 5a, how many replicates, and what is the bar represent for?
- Fig 6d and Fig S7, how many replicates for the qPCR?
- Background of Fig. S2, S7, S10 should be white.
- The font size should be uniform, for example, the font in y axis of Fig 4b and Fig 4e are different.
- Please provide the accessions for all the raw data for DNA methylation, ChIP-seq, Illumina, PacBio, Hi-C, GWAS etc.

Reviewer #3:

Remarks to the Author:

Peony has several prominent features, including a giga genome with giga-chromosomes, double flowers that are more attractive and valuable for ornamental crops, and >90% of polyunsaturated fatty acids (PUFA) in seed oil. The authors applied genomic and metabolomic technologies to investigate these intriguing biological questions. They discovered explosive LTR retrotransposon expansion caused massive genome size increase without any recent whole genome duplication after the triplication event shared by all eudicots. The massive amount of histone proteins needed for this giga-genome are supplied by the expansion of gene families encoding the five types of histones, particularly genes encoding H2A.W and H3.1, which promote chromatin condensation, DNA replication, and chromosome assembly. The petaloid-stamens contributing to double flowers are caused by ectopic expression and class A gene AP1 and reduced gene expression of class C gene AG. GWAS analysis of 448 peony accessions on 35 traits related to fatty acid biosynthesis revealed a cluster of SAD and FAD genes contributing to PUFA biosynthesis. These findings substantially enhanced our understanding of genomic and molecular bases of these biological features. The knowledge and genomic resources gained will benefit plant research community.

Minor changes:

Line 37 - 38: "...but the species still closes to extinction" – ...but the species is close to extinction

Line 85: It would be better to start a new paragraph describing seed oil content.

Lines 136 – 137: Only one sentence stating the chloroplast and mitochondrial genomes were assembled. Although the chloroplast genomes are highly conserved, mitochondrial genomes are variable among flowering plants. I would suggest using a paragraph to describe these two organelle genomes. Because genome evolution is a major part of this manuscript, it is necessary to assess whether organelle genomes evolved at a different pace.

Lines 184: "...is tens of times larger..." – It is better to use actual number here.

Line 197: A new paragraph could start from "To explore protein-DNA interactions..."

Line 270: "...and assembled in endoplasmic reticulum..."- ...and assembled in smooth endoplasmic reticulum...

Reviewer #4:

Remarks to the Author:

The paper deals with an interesting phenomenon of genome structure in the peony, which has a relatively large genome but only five chromosomes. I am pleasantly surprised by the set of methodological arsenal used to unravel the nature of giga-chromosomes and have no doubt that the results deserve the attention of the scientific audience. On the other hand, I'm not too happy with some of the evolutionary implications drawn - they are possible, but I don't think you have enough evidence to support them. Meanwhile, some of the claims (especially in the discussion) strike me as outside the scope of the paper, i.e. unnecessary given that the findings themselves are solid enough. See my specific comments on each section of the text.

Introduction

p. 3, l. 59-62 – The range of genome sizes in Angiosperms should be corrected - the smallest genome size is in *Genlisea aurea* (instead of *G. margaretae* - see Fleischmann et al. 2014 <https://doi.org/10.1093/aob/mcu189>) and the largest is not in *Viscum album* (its genome size in 1C-

value, i.e. in the same value as *Genlisea*, is "only" 102.9 pg - Zonneveld 2010, <https://www.hindawi.com/journals/jb/2010/527357/>), but for *Paris japonica* - 1C = 149.2 Gb (Pellicer et al. 2010 - <https://doi.org/10.1111/j.1095-8339.2010.01072.x>). Both values must be in the same form and units (i.e., 1C value in Mb or Gb).

Results

p. 8-9, l. 169-175 + Table S24 – I recommend unifying the terminology of LTR elements - in the text you use the subtype Gypsy/del, while in table S24 you use the subtype Gypsy/Tekay

p. 9, l. 176-178 – How was the dating of LTR bursts determined? I don't understand the suggested coincidence with volcanic eruptions (see also comments to Discussion).

p. 9, l. 181-183 – I don't understand the meaning - the enzymes for synthesis of del were 8.25 times more than what? I assume "than for synthesis of other LTRs", but the meaning is different.

p. 13, l. 283-285 – Interpretation of results belongs to Discussion section

p. 14, l. 294-304 – I don't understand the placement of this paragraph in the results - it should be in the introduction or discussion. The results section should be dedicated to the results only. In addition, please omit the adjective "beautiful" if you are talking about petals. The whole Petaloid-stamen results section is actually written differently than the other sections, combining results with discussion, which is peculiar in the overall context of the other sections.

discrepancy between Table S23 and S24 – I don't understand the calculation of the proportion of the genome made up of LTR elements. In Table S23 you state that LTR retrotransposons in *P. ostii* make up ~43% of its genome. At the same time, in Table S24, you attribute only ~12% to the proportion of LTRs in the genome.

Discussion

p. 17, l. 339 – How does your statement "*P. ostii* has the largest chromosome size known in plants sequenced to date" relate to the cited Hidalgo et al. 2017 paper? As far as I know, the paper deals with huge genomes in plants (more than 100 Gb), which is far beyond the relatively common genome size of *Paeonia* (~12 Gb). Moreover, the paper does not discuss chromosome size in sequenced plants at all.

p. 17, l. 344-348 – I found no evidence in the paper that provides a basis for such a strong statement. Could you explain, not only in the discussion but also in the M&M and Results sections, on what basis you date the expansion of LTR to the recent 1-2 My?

p. 18, l. 365-368 – You repeatedly talk about volcanic events in the last two million years and their relation to adaptive evolution in *Paeonia*, but you never specify what you mean by "volcanic events" and which ones are essential for the evolution of giga-chromosomes. Are you sure you know what you're talking about? Do you really mean volcanic, i.e. volcanic eruptions?

p. 18, l. 368-375 – I'm not too familiar with the way you're conducting the discussion. Have you analyzed the seed set in relation to giga-chromosomes or genome size? I don't think so, so why are you going down such a speculative path that is completely out of line with the very interesting findings you have made? Why you repeatedly use the term giga-genome? The size of the genome of *Paeonia ostii* is not gigantic, its genome is only one of the larger ones among plants.

Methods

Take care to write plant names in strict binomial form - *Vitis vinifera* instead of *Vitisvinifera*, *Malus domestica* instead of *Malusdomestica*, etc.

Figures

Fig. 2 – As all images should be easy to understand (self-explanatory), abbreviated plant names should be explained in the caption.

Fig. 5c – The meaning of the red asterisk and the triangle in phylogeny is not clear.

Tables

In several cases (Table S19, S24) you used only abbreviated plant names (only the first letter of the genus name), which is difficult to understand.

REVIEWER COMMENTS

Thanks for the comments. The authors appreciate all the valuable comments from the reviewer. All authors worked hard to revise the manuscript. We did more analyses according to the reviewer's suggestions, and revised the manuscript accordingly with changes highlighted in yellow. The corresponding responses were listed below point-by-point.

Reviewer #1 (Remarks to the Author):

This study reported a large genome of *P. ostia* based on huge amount of PacBio and Hi-C data.

1. Although the mapping rate of RNA-seq is a little low, the quality of assembly is overall good since the genome is very large. How about the heterozygosity? I did not see the information. If it is high, the assembly would be challenge and the genome quality will be affected.

Response:

The heterozygosity was estimated to be 1.5% by GenomeScope (as shown in Fig. S1b). The resolution in the combined figure was reduced in the uploaded Word manuscript, it is a bit difficult for the reviewer to distinguish the numbers. Sorry for the inconvenience. The publisher would help us replace the high-quality version with higher resolution when the manuscript is available online. Meanwhile, in order to express the content more comprehensively, the authors added more interpretations in

the Fig. S1b figure legend as ‘The heterozygosity was estimated to be 1.5%’. As pointed out by the reviewer, the high heterozygosity is indeed a great challenge to genome assembly. Therefore, the authors applied Illumina short-read sequencing data, PacBio long-read sequencing data, and Hi-C data with a variety of algorithms to improve the quality of assembly.

2. The analysis on fatty acid synthesis has been done in many species and the pathway and key genes are well -known. Thus, this part provides limited novelty and progress.

Response:

The authors agree that the fatty acid synthesis pathway and key genes are well-known, while our work and many previous studies have indeed proved that the fatty acid synthesis pathway and key genes are conservative in most species, but still there are great differences in the composition and content of fatty acids in seeds of tree peony

and many species. For example, tree peony has a high content of ALA, while the content of ALA in cereal is low, and the mechanism behind this is unclear. Our work has made quiet progress. Through a combination of multi-omic data (genomics, transcriptomics, lipidomics), we not only explained the mechanism of efficient accumulation of high PUFA in tree peony, but also carried out the evolutionary analysis of genes encoding PUFA desaturase comprising of representative groups within the scope of land plants. Three functional *FAD3* genes were verified by wet lab experiment in this work and the authors would initiate more verifications in the upcoming future, although it is quite challenging.

3. One major concern is about fatty acid profiling and GWAS analysis on fatty acids. First, for fatty acids in seed oil, there are just a few major types. Authors seemed to search the NIST database and identified a lot of minor ones. I believe some fatty acid species (such as C14:0, C17:1 and so one) is wrong by searching the database (the software tells wrong prediction). I suggest authors just analyze major species: C16:0, C18:0, C18:1, C18:2, C18:3 and so on. Second, authors have not presented the distribution of these fatty acid traits, we can not just the quality of data. Third, author should do GWAS for each fatty acid. It seems that authors have not done this for major ones. Is it because the fatty acid composition or content is not accurate? Then, they may do GWAS for the ratio of fatty acids. Fourth, authors showed SAD and FAD2 in the loci identified by GWAS. How about FAD3 in any loci? Since ALA directly synthesized by FAD3 is the focus of

this study. Anyway, this part is very unsatisfactory.

Response:

Thanks for the comments. The authors made many corrections in this section as suggested by the reviewer. More information was supplied to make the presentation more concise and accurate.

1) We deleted the minor fatty acid species (such as c14:0, c17:1 and others), while focusing on several major species as pointed out by the reviewer. Please see the revised supplementary table 29, 30, and 32.

Trait Name	Short Name
Content of alpha-linolenic acid (C18:3^{Δ9,12,15})	ALA
Content of linoleic acid (C18:2^{Δ9,12})	LA
Content of oleic acid (C18:1^{Δ9})	OA
Content of steric acid (C18:0)	SA
Content of palmitic acid (C16:0)	PA
Content of multiple fatty acids (PA+SA+OA+LA+ALA)	MFA
Content of total fatty acids	TFA
Ratio of C18:2^{Δ9,12}/C18:3^{Δ9,12,15}	RLAALA
C18:1^{Δ9}/C18:2^{Δ9,12}	C1819C182912
C18:0/C18:1^{Δ9}	C180C1819
C16:0/C18:0	C160C180
Content of Unsaturated fatty acids (OA+LA+ALA)	UFA
Content of Saturated fatty acids (PA+SA)	SFA
Ratio of Unsaturated fatty acids/ Saturated fatty acids	RUS

2) We have supplied more information on the description of methods (on Page 40-42 highlighted in yellow). And the distributions of these fatty acid traits were analyzed as suggested by the reviewer.

3) We believe that the experimental conditions in this study were unified. All the plant materials were cultivated in Shanghai Chenshan Botanic Garden and collected in four consecutive years. The authors made cautious efforts and laborious work during sample

collection and preparation. While the method for fatty acid quantification is rigorous, please see the revised descriptions in the method section. The reliability of our data is credible.

4) The authors spent great efforts to obtain phenotypic data and performed SLAF-seq. As the reviewer guessed, we have conducted SLAF-seq data analysis, but *FAD3* was not found in the GWAS results which was not in line with our expectation. It is quite regretful, but this result comes from actual data and authentic statistical analysis. For instance, in the following graph a unique locus about *FAD3* was just below the threshold for statistical significance. Considering the impact of temperature fluctuation between years and other factors, we used the four-year average value and several GWAS statistical models to get the optimal results. As expected, we found many loci, especially the upstream genes for ALA synthesis, such as *SAD* and *FAD2*. Transcriptome sequencing showed that these two genes were highly expressed during seed development. Therefore, it is confident that these genes are important in the oil synthesis of tree peony seeds.

4. The method for fatty acid was not well described. How was the methylation done?

Temperature and methylation time? Which fatty acid was used as standard?

Internal or not? How about fatty acids analysis for the GWAS population?

Authors failed to provide any information about sampling, replicates and so on.

Response:

More detailed information on experimental materials and methods have been supplied in the method section. Now it reads,

In total, 448 individuals of *P. ostii* from five areas including Luoyang, Tongling, Bozhuo, Hezi and Shaoyang of China were introduced in the special Paeoniaceae nursery at Shanghai Chenshan Botanical Garden (31°4'52"N, 121°10'14"E) since 2014. Over four growing seasons from 2016 to 2019, the budded flowers of each of 448 plants were hand-pollinated with the pollen collected from the same *P. ostii* plant and the seeds were obtained per plant annually. Fatty acid contents of individual plants were measured annually and the protocols were following our previous lab publication (Yu et al. [Scientific Reports, 2016]). Fatty acid samples were prepared according to the

procedures described by Bligh and Dyer (Bligh, E. G. & Dyer, W. J. A rapid method of total lipid extraction and purification. *Can. J Biochem. Physiol.* 37, 911-917 (1959)). The concentrated seed lipids prepared above were re-dissolved in 2 ml H₂SO₄ methanol solution (4% H₂SO₄). After charging with nitrogen gas, the sample was vortexed for 1 min and placed in 90 °C water bath for 1 h. Then the sample was mixed by vortexing after the addition of 1 ml ddH₂O and 1 ml hexane, followed by centrifugation at 4000 g for 15 min. The supernatant was transferred to a new tube, concentrated by bubbling nitrogen and stored in 4 °C for GC-MS analysis. Additionally, 20 µl nonadecanoic acid (50 mg/mL in hexane) was used as the internal standard for each sample. The FA methyl esters were subjected to GC-MS (GC7890/MS5975, Agilent) on a HP-88 capillary column (60 m × 0.25 mm, 0.2 µm, Agilent). The column temperature was held at 70 °C for 1 min, increased to 210 °C at 10 °C/min for 0 min and then to 220 °C at 10 °C/min for 0 min, and subsequently to 235 °C at 10 °C/min for 8 min. The injector temperature was set at 250 °C for split injection at a split ratio of 5:1. The injection volume was 1 µl. Helium was used as the carrier gas at a flow rate of 1 ml/min, and the ionisation potential of the mass-selective detector was 70 eV. FA identification was achieved through a mass spectrum database search (NIST MS Search 2.0) and co-eluted with the 37-component FAME Mix (Sigma, USA). A standard curve method with an internal standard was used as the quantitative approach to construct three calibration plots of internal standard peak-area ratio versus standard concentration, as determined by the least squares method. The five major FAs in each sample were quantified in absolute terms using the linear regression of their corresponding standard, while the minor FAs

were measured using methyl nonadecanoic acid as the internal standard. FAMES were expressed as milligrams per gram DW of sample. All samples were analyzed in triplicate. The statistical analysis method was used for data processing according to the procedures described by Dunn and Clark (Dunn, O. J. & Clark, V. A. Applied statistics: analysis of variance and regression. J. Educa. Statis. 15(2), 175-178 (1990).). FA contents and percentages were tested by One-way ANOVA analysis ($p < 0.05$) and comparisons between means were performed with Tukey's test.

5. Line 89 to 90, fatty acid synthesis is conserved. We know FAD2 and FAD3 are key genes determining the content of linoleic acid and α -linolenic acid (ALA). It is certainly the same in *P. ostia*. I believe it is very likely previous studies have some information on this (such as gene cloning, transcriptome analysis). Authors should check if any works have studied linoleic acid and α -linolenic acid synthesis.

Response:

Thanks for the comments. Fatty acid synthesis is conserved in various plants, and as pointed out by the reviewer, indeed several previous works (listed below) studied linoleic acid and α -linolenic acid synthesis in tree peony.

These studies have predicted that the biosynthesis of UFAs in tree peony seeds is mainly caused by the high expression of *SAD*, ω -6 and ω -3 *FAD* genes via comparative transcriptome and proteomic analyses or *FAD3* gene colonization and functional test. However, without genomic information, several ω -3 *FADs* were annotated as *FAD3*, *FAD7*, or *FAD8* in different databases, relating to the different identification of ω -3 *FAD*

in the diverse alignment database. There were great differences in the identification and copy number of omega-3 *FAD* genes, which were not concise and rigorous. For example, Li et al. reported eight omega-3 *FADs* (one for *FAD3*, five for *FAD7* and two for *FAD8*), most of the *FAD7/8* genes were incorrectly annotated, especially those so-called ‘*FAD8*’ reported in this paper were actually *FAD3*. In another Li et al. article, the cloned gene *PoFAD8* (MH049427.1) should also be annotated as *FAD3*. What is more, the *FAD3s* (CL4770.Contig4_F and CL4770.Contig5_F, TAIR ID: AT5G05580) reported in Wang et al. should be *FAD8* (2021, 11:616338). These mistakes accrued in several papers, thus we did not cite these papers. Based on our genomic and transcriptomic data, the results confirmed the *FAD3* gene copy number and accurately identified the duplication events (*FAD3* and *FAD7/8*) of the omega-3 *FAD* gene family in angiosperms. We not only comprehensively analyzed all the genes in the conserved FA synthesis pathway of tree peony, but also showed that at least one copy of the genes at each key node in the synthesis pathway is highly expressed. Several genes acting on key nodes, including *SAD* (13 genes), *FAD2* (four genes), *ketoacyl-ACP synthase 1* (*KASI*, three genes), and *FAD3* (four genes) showed expansion and high expression, which might help to maintain the high ALAs content in tree peony.

1. Li, S.S. *et al.* Fatty acid composition of developing tree peony (*Paeonia* section Moutan DC.) seeds and transcriptome analysis during seed development. *BMC Genomics* **16**, 208 (2015).

2. Xiu, Y. *et al.* Oil biosynthesis and transcriptome profiles in developing endosperm

and oil characteristic analyses in *Paeonia ostii* var. *lishizhenii*. *J Plant Physiol* **228**, 121-133 (2018).

3. Li, L. *et al.* Characterization of genes encoding omega-6 desaturase *PoFAD2* and *PoFAD6*, and omega-3 desaturase *PoFAD3* for ALA accumulation in developing seeds of oil crop *Paeonia ostii* var. *lishizhenii*. *Plant Sci* **312**, 111029 (2021).

4. Yu, S.Y. *et al.* Transcriptomic analysis of alpha-linolenic acid content and biosynthesis in *Paeonia ostii* fruits and seeds. *BMC Genomics* **22**, 297 (2021).

5. Wang, X. *et al.* Integrated analysis of transcriptomic and proteomic data from tree peony (*P. ostii*) seeds reveals key developmental stages and candidate genes related to oil biosynthesis and fatty acid metabolism. *Hortic Res* **6**, 111 (2019).

6. Wang, M. *et al.* Interspecific variation in the unsaturation level of seed oils were associated with the expression pattern shifts of duplicated desaturase genes and the potential role of other regulatory genes. *Front Plant Sci* **11**, 616338 (2020).

6. Line 269 to 292, it is true ALA can be synthesized in plastid by FAD7/8 (in green tissue such as leaves, do not contribute to ALA in seed oil) and in ER by FAD3 in seed (contribute to ALA in seed oil). Authors may not be quite clear about this and some of the writing is misleading. If they want to address ALA in seed oil, they should focus on FAD3. If they want to talk about ALA in leaves and other green tissues (ALA is essential for membrane lipids and related to many biological functions), then FAD7/8 should be analyzed. In this case, Fig 5 may focus on ALA in seed oil and FAD6, FAD7/8, may not be shown in Fig 5.

Response:

Thanks for pointing this out. The context was corrected as suggested by the reviewer.

Now it reads, “both plastid-oriented *FAD7/8* (in green tissue such as leaves, do not contribute to ALA in seed oil) and ER - oriented *FAD3* in seed (contribute to ALA in seed oil).”

It is much appreciated that the reviewer’s opinions on the modification of Fig. 5 are very valuable, but retaining *FAD6* and *FAD7/8* in this paragraph might give peer experts a more comprehensive picture of key genes in the ALA synthesis pathway, as it is broad interest for the researchers to look at the whole information at one time. Here we are more interested in ALA in seed oil, therefore we focused more on *FAD3* with bioinformatics analysis and experimental validation (while *FAD7/8* was just mentioned without too many words). Also as suggested by the reviewer, in another study we have reported the comparison of ALA synthesis in different tissues (fruit, seed coat and kernel) (Yu, S.Y. *et al.* Transcriptomic analysis of alpha-linolenic acid content and biosynthesis in *Paeonia ostii* fruits and seeds. *BMC Genomics* **22**, 297 (2021)).

We made several correction related to this comments in Introduction, Results and Conclusion part to make our manuscript more concise.

7. For results in Supplemental Fig. 8, authors need to calculate the composition or content of produced ALA. Not just showing a few pictures.

Response:

Thanks for the comments. A supplemental table was supplied to show the contents of fatty acids in transgenic yeast grown at different temperature with linoleic acid (LA) added as a substrate at 72 h. Please see supplemental table 36.

Yeast	T/°C	Fatty acid						% Conversion
		16:0	16:1 ^{Δ9}	18:0	18:1 ^{Δ9}	18:2 ^{Δ9,12}	18:3 ^{Δ9,12,15}	
pY31 (PoFAD3_1)	28	19.30±0.09	16.46±0.14	6.39±0.14	11.21±0.06	39.73±0.06	–	–
	25	10.29±0.06	39.28±0.22	6.88±0.02	32.33±0.21	7.04±0.48	–	–
	20	10.39±0.50	43.84±0.58	5.82±0.26	29.52±0.23	5.06±0.23	1.20±0.02	19.24±0.83
	15	10.98±0.60	49.63±0.68	3.91±0.22	27.25±0.43	3.05±0.25	2.86±0.25	48.38±3.85
	10	14.38±0.28	41.29±0.11	4.42±0.15	22.99±0.36	11.15±0.19	0.93±0.04	7.71±0.34
	5	15.02±0.17	41.72±0.04	4.23±0.06	20.16±0.01	13.34±0.22	0.92±0.03	6.47±0.11
Py33(PoFAD3_3)	28	23.21±0.38	5.51±0.21	7.23±0.35	3.55±0.27	51.95±0.13	–	–
	25	9.93±0.05	38.49±0.16	6.25±0.18	32.52±0.29	8.30±0.28	–	–
	20	8.52±0.05	43.87±0.56	5.54±0.09	32.05±0.39	4.78±0.08	0.61±0.08	11.33±1.53
	15	11.54±0.25	49.27±1.49	3.54±0.21	28.52±0.76	2.94±0.05	1.51±0.05	34.04±0.71
	10	14.47±0.42	42.48±1.18	4.37±0.32	22.32±1.13	11.16±0.32	0.76±0.01	6.36±0.18
	5	16.43±0.19	39.13±0.37	4.55±0.28	20.97±0.12	14.10±0.19	0.46±0.04	3.16±0.25
Py34(PoFAD3_4)	28	21.08±0.29	10.15±0.15	6.66±0.32	7.19±0.21	49.42±1.00	–	–
	25	9.89±0.17	38.14±0.73	6.45±0.34	33.02±0.38	7.03±0.30	1.31±0.03	15.73±0.65
	20	9.15±0.27	43.26±0.65	5.56±0.29	31.30±0.47	2.47±0.15	3.47±0.06	58.47±1.09
	15	12.33±0.32	48.69±0.82	3.67±0.17	27.08±0.62	2.33±0.06	3.01±0.08	56.39±0.27
	10	14.67±0.46	39.27±0.71	5.07±0.12	20.95±0.16	13.77±0.26	1.11±0.05	7.45±0.42
	5	13.41±0.98	41.86±0.16	3.97±0.24	23.38±0.21	11.97±0.65	1.26±0.08	9.51±0.13

Some minor issues:

How many SNPs were used for GWAS? Authors should provide such key information.

Response:

In total, 1,022,648 SPNs were used for GWAS analysis. Please see the table below.

Chromosome	SNPPercentage	SNPNumber
Chr01	0.2139	218,785
Chr02	0.1845	188,662
Chr03	0.1489	152,290
Chr04	0.2224	227,439
Chr05	0.2084	213,113
unchr	0.0219	22,359
		1,022,648

The method for yeast experiment was not provided. And so fatty acid analysis for yeast.

Response:

Thanks for the comments. More information was supplied in the method section. Now it reads,

Fatty acid linolate desaturases (*FAD3* and *FAD7/8*) are key enzymes that catalyze the production of ALA, C18:3 from LA, C18:2, especial *FAD3* in peony seed development. In common, *FAD7/8* is located in plastid and *FAD3* is located in endoplasmic reticulum (ER). The fused expression vector with green fluorescent protein (GFP) was transferred into *Agrobacterium tumefaciens* using the freeze-thaw method, and simultaneously infected *Nicotiana benthamiana* leaves.

Based on the Gateway technology, the expression vectors (pDonr207 and Pyes-DEST52 donated by Prof. Qin Zhao, Shanghai Chenshan Botanical Garden) of *FAD3* genes (*FAD3_1*, *FAD3_2*, *FAD3_4*) were constructed, and transferred into *Saccharomyces cerevisiae* INVSc1 (Ura⁻ defects, donated by Prof. Zhi-gang Zhou, Shanghai Ocean University). The transformants (pY31 with *FAD3_1*, pY33 with *FAD3_3* and pY34 with *FAD3_4*) were selected on SC minimal medium that was deficient in uracil (SC-U) and containing 2% glucose as the sole carbon source. The wild type and transgenic yeasts were fed with LA, and 2% galactose was added as an inducer. The fatty acid profiles of yeast were detected by gas chromatography-mass spectrometer (GC-MS) (detailed methods referred to Yu et al., Scientific Reports, 2016).

Line 717, why did authors describe qRT-PCR here?

Response:

qRT-PCR analysis was used to validate the expression of key genes in ALA biosynthesis. More detailed information was supplied in the method section. Now it reads, Eight key genes (*FAD3_4*, *FAD7/8*, *SAD_3*, *FAD2_1*, *FAD6*, *CALO_5*, *STERO_4* and *OLE_1*) associated with FA biosynthesis and TAG assembly, in particular those relating to ALA bio-synthesis, were selected for qRT-PCR validation. The expression levels of these genes were quantified in the seeds at 35, 49, 63, 77, 91 and 119 DAF (12 samples in total). qRT-PCR with three replicates for each gene was performed on the ABI StepOnePlus platform with *actin* as the internal standard. Total RNA was isolated from each tissue at various developmental stages using RNA Exaction Kits (E.Z.N.A. HP Plant RNA Kit, Omega Bio-Tek). First-strand cDNA was prepared from 1 µg of total RNA per sample using a FastKing RT Kit with gDNase (Tiangen). PCRs were performed on an ABI StepOnePlus® Real-Time PCR System (Applied Biosystems), following the manufacturer's instructions. Each reaction mixture (20 µl) contains 10 µl of TB Green Premix Ex Taq II (Tli RNaseH Plus) (Takara), 0.8 µl of each primer (10 µM), 0.3 µl of cDNA template (1 µg), and 8.1 µl of RNase-free water. PCRs for each gene were performed in triplicate, with the following thermal cycling conditions: 95°C for 30 s; 40 cycles of 95°C for 5 s and 64°C for 30 s; and 95°C for 15 s. Primer specificity was confirmed by melting curve analysis. The relative expression levels of the tested genes were calculated using the $2^{-\Delta\Delta C_t}$ method, using the *actin* genes as internal controls.

Line 75, value. It is also the parental

Response:

Thanks for pointing this out. The context was corrected as suggested by the reviewer.

Line 83, species of *P. ostia*. Genomic information would

Response:

Thanks for pointing this out. The context was corrected as suggested by the reviewer.

Line 85, the information is wrong. It should be “a high proportion (more than 90%) of unsaturated fatty acids. Authors need to understand the difference between unsaturated fatty acids and polyunsaturated fatty acids (PUFAs).

Response:

Thanks for pointing this out. The context was corrected as suggested by the reviewer.

Line 204, and giga-genome.

Response:

Thanks for pointing this out. The context was corrected as suggested by the reviewer.

Line 222, with other plant species. Similarly, the expansion of H2A. Authors need to avoid the using of “;” in the manuscript

Response:

Thanks for pointing this out. The context was corrected as suggested by the reviewer.

All the “;” was deleted in the manuscript.

Line 267, high ALA content

Response:

Thanks for pointing this out. The context was corrected as suggested by the reviewer.

Line 514, K-mer analysis

Response:

Thanks for pointing this out. The context was corrected as suggested by the reviewer.

Line 706, P. ostia were

Response:

Thanks for pointing this out. The context was corrected as suggested by the reviewer.

Reviewer #2 (Remarks to the Author):

In this manuscript titled “Genomic basis and evolutionary adaptation of giga-chromosome and giga-genome of tree peony *Paeonia ostia*”, Yuan and colleagues present the genome of tree peony *Paeonia ostii*, with the largest chromosome in all sequenced plants. The authors also identified genes for PUFA biosynthesis by GWAS. Overall, this manuscript is interesting and well-written. The genome assembly and GWAS data could be very helpful for the plant community.

One interesting feature of the Paeonia ostia genome is its large chromosome. The author proposed that the expansion of histone variants contribute to the giga chromosome size. While the result is preliminary and could be strengthened by performing a statistic analysis to examine if the association between number of histone variants (Table S26) and chromosome size is significant or not.

Response:

Thanks a lot for the valuable comments. The reviewer had insightful knowledge on our project. We then performed a statistical analysis to check the association between number of histone variants and chromosome size as suggested by the reviewer. The detailed information was updated in the revised Table S26. We tried several statistical formulars, there was no significant linear relevance between total histones and chromosomes length. But still, we added these data in Table S26 for further researchers, hopefully this would help them to explore more information if they are working on large chromosomes and genomes.

Species	H1	H2A	H2B	H3	H4	Total	Genome size (Gb)	Chr number (2n)	Chr number (n)	Chr length (Mb)	Ploidy
Gymnosperms											
Ginkgo biloba	4	24	15	17	16	76	9.87	24	12	822.50	2
Pinus taeda	2	15	12	5	16	50	20.15	24	12	1,679.17	2
Basal Angiosperms											
Amborella trichopoda	3	8	5	9	9	34	0.748	26	13	57.54	2
Nelumbo nucifera	6	15	8	18	10	57	0.783	16	8	97.88	2
Monocots											
Brachypodium distachyon	3	13	12	15	10	53	0.272	10	5	54.40	2
Oryza brachyantha	3	12	10	8	8	41	0.261	24	12	21.75	2
Setaria italica	4	14	11	16	9	54	0.423	18	9	47.00	2
Oryza sativa	4	13	11	14	10	52	0.466	24	12	38.83	2
Musa acuminata	7	20	10	16	15	68	0.523	22	11	47.55	2
Sorghum bicolor	4	23	13	14	11	65	0.625	20	10	62.50	2
Elaeis guineensis	7	18	10	14	10	59	1.8	32	16	112.50	2
Zea mays	6	33	15	14	16	84	2.2	20	10	220.00	2
Aegilops tauschii	5	43	34	14	30	126	4.3	14	7	614.29	2
Hordeum vulgare	3	13	5	9	1	31	4.79	14	7	684.29	2
Dicots											
Arabidopsis thaliana	3	13	11	13	8	48	0.125	10	5	25.00	2
Capsella rubella	3	14	9	12	7	45	0.134	16	8	16.75	2
Arabidopsis lyrata	3	22	13	15	9	62	0.207	16	8	25.88	2
Fragaria vesca	4	10	8	14	8	44	0.24	14	7	34.29	2
Prunus mume	3	12	7	8	6	36	0.28	16	8	35.00	2
Eutrema salsugineum	3	14	8	11	8	44	0.241	14	7	34.43	2
Prunus persica	3	13	6	8	7	37	0.228	16	8	28.50	2
Cucumis sativus	3	9	7	6	7	32	0.289	14	7	41.29	2
Tarenaya hassleriana	6	19	12	14	9	60	0.29	20	10	29.00	2
Citrus clementina	3	10	10	9	4	36	0.301	18	9	33.44	2
Jatropha curcas	6	14	7	10	6	43	0.379	22	11	34.45	2
Citrus sinensis	3	10	5	8	4	30	0.32	18	9	35.56	2
Ricinus communis	5	12	6	11	6	40	0.336	20	10	33.60	2
Cucumis melo	4	10	7	7	7	35	0.375	24	12	31.25	2
Theobroma cacao	4	11	6	9	8	38	0.327	20	10	32.70	2
Vitis vinifera	5	15	5	9	6	40	0.487	38	19	25.63	2
Sesamum indicum	4	16	9	12	14	55	0.274	26	13	21.08	2
Medicago truncatula	6	27	20	33	15	101	0.393	16	8	49.13	2
Populus trichocarpa	8	19	12	18	14	71	0.392	38	19	20.63	2
Beta vulgaris	3	10	7	15	8	43	0.394	18	9	43.78	2
Vigna radiata	4	10	9	9	7	39	0.475	22	11	43.18	2
Phaseolus vulgaris	4	14	10	11	12	51	0.549	22	11	49.91	2
Eucalyptus grandis	3	9	7	9	7	35	0.46	22	11	41.82	2
Cicer arietinum	4	11	12	12	8	47	0.532	16	8	66.50	2
Malus domestica	10	27	11	22	15	85	0.598	34	17	35.18	2
Gossypium raimondii	5	22	13	16	14	70	0.775	26	13	59.62	2
Solanum lycopersicum	4	16	13	14	9	56	0.76	24	12	63.33	2
Glycine max	8	24	14	21	19	86	0.95	40	20	47.50	2
Solanum pennellii	4	17	13	15	10	59	1.2	24	12	100.00	2
Arachis duranensis	3	13	8	11	7	42	1.25	20	10	125.00	2
Arachis ipaensis	4	12	9	13	9	47	1.56	20	10	156.00	2
Paeonia ostii	14	43	54	46	51	208	12.28	10	5	2,456.00	2

For the genome annotation, *Paeonia suffruticosa* (PMID: 32551041) contains 34,854 protein coding genes, which is much less than 73,177 protein coding gene that reported in this study. I suggest the authors investigate this big differences of the number between the two related tree peony species. And how many of these protein coding genes are supported by RNA-seq?

Response:

Thanks for the comments. For *Paeonia suffruticosa*, a total of 35,687 genes were annotated, while 34,854 of them were functionally annotated. Considering the giga-

genome of tree peony, we followed the idea on annotation method from barley, which is also a large genome (Mascher M, Gundlach H, Himmelbach A, et al. A chromosome conformation capture ordered sequence of the barley genome[J]. Nature, 2017, 544(7651): 427-433.). In our project, stringent confidence classification was applied to all predicted genes to discriminate loci representing high-confidence (HC) protein-coding genes and less reliable low-confidence (LC) genes, which potentially consisted of gene fragments, putative pseudogenes or TE-related genes. The genes were classified by the following conditions: (1) genes with significant high sequence homology (BLASTN e-value <1e-10) to the repeat library and TEs were considered LC genes. (2) Peptide sequences of peony genes were compared against the protein data sets of five homologous species (Arabidopsis, apple, *Kalanchoe fedtschenkoi*, papaya and grape) and the SwissProt database (<https://www.uniprot.org/>) using BLASTP (e-value <1e-10). The best-matched reference protein was selected as a template sequence. Genes were defined as HC genes if they had a significant BLASTP (e-value <1e-10) hit to reference proteins and over 60% identity to the respective template sequence. More details could be found in the method section. In total, serial annotation identified 73,177 protein-coding gene models (including 54,451 high-confidence genes anchored to the chromosome) (please see the table below).

Final gene set	73,177
High confidence genes	55,998
Low confidence genes	17,179
High confidence genes anchored to the chromosome	54,451
Low confidence genes anchored to the chromosome	16,244

In order to compare the difference between tree peony gene set and *Paeonia suffruticosa*, we checked the homogeneity of all 70,695 genes (HC and LC genes anchored to the chromosome) of tree peony to *Paeonia suffruticosa* genome through Exonerate. The data showed that 69,244 (97.95%) of tree peony genes could be aligned to *P. suffruticosa* genome, while among them 67,263 genes (95.15%) were with coverage>50%. This result indicated that the gene we annotated also existed in *Paeonia suffruticosa* genome, but *Paeonia suffruticosa* was not well annotated.

In addition, a total of 43,746 (80.34%) genes in our HC genes could be supported by NGS RNA-seq or Iso-Seq data. And 97% (52,731) of the HC genes could be identified in commonly used protein function databases (Swissprot, TrEMBL, KEGG, GO, InterPro, KOG, NR). All these data support the high credibility of our assembly and annotation.

I am also amazed by the extremely large number (330,511) pseudogenes identified in *P. ostii* genome. Could the author do some analysis to find some feature/pattern for these pseudogenes, to explain the largest number of pseudogenes to date.

Response: The large number of pseudogenes of *P. ostii* was mainly originated from LC genes (253,139 pseudogenes in LC genes), and only 77,372 pseudogenes were from HC gene. It showed similar patten in wheat D genome (Zhao G, Zou C, Li K, et al. The *Aegilops tauschii* genome reveals multiple impacts of transposons[J]. Nature Plants,

2017, 3(12): 946-955.), a total of 267,546 pseudogenes were predicted.

Following the idea from the ‘wheat D genome paper’ published in *Nature plants*, we checked whether the pseudogenes were related to TE. By comparing the number of introns of the pseudogenes and the ancestor genes, we investigated whether introns had been lost in the pseudogenes. If it is lost, it is considered that the pseudogenes were originated by retrotransposition. As a result, 81,473 processed pseudogenes were identified, accounting for 24.65% of the pseudogenes. It means at least 24.65% of the genes were mediated by TE. In addition, according to the reviewer's suggestion, in order to view the relationship between pseudogenes and TE in a more detailed and intuitive way, we further analyzed the distribution pattern of TE in the upstream and downstream regions of pseudogenes. The distribution density map of TE within 20Kb upstream and downstream of all pseudogenes were shown in below.

Minor points

-The circos plot in Figure 1 is common for genome paper, but I think this figure didn't convey much information for the audience. I suggest reducing size of this figure, and not necessary to label so many numbers for genome coordinates.

Response:

Thanks for the comments. Figure 1 was modified and updated. Only the key information was left in the figure.

-The replicates number of sequencing library should be mentioned in the method

for all related experiments.

Response:

Thanks for the comments. The DNA libraries for sequencing were listed in Table S1, S2 and S3. All the RNA libraries had three replicates, and we added this information in the corresponding method sections.

Table S1. Statistics of Illumina sequencing data.

Clean Reads Length(bp)	Clean Data(bp)	Depth(X)
100	74,390,370,000	6.19
125	257,499,996,500	21.44
125	246,291,814,500	20.51
125	165,689,612,000	13.79
230	469,803,490,302	39.12
230	451,138,298,378	37.56
49	305,980,362,212	25.48
49	398,447,818,076	33.18
49	195,335,363,902	16.26
49	241,475,980,466	20.1
49	168,412,894,650	14.02
	2,974,466,000,986	247

Table S2. Statistics of PacBio SMRT sequencing data.

PacBio SMRT seq	Total Bases (Gb)	Average read length (bp)	N50 length	Quality	Depth (X)
RSII	277.64	9,122	11,882	85%	23.12
Sequel	366.03	11,114	17,315	85%	30.48
Total	643.67	10,118	14,598	85%	53.6

Table S3. Statistics of Hi-C sequencing data.

Library	Reads	Bases(Gb)
MuDan1	493,949,008	148.18

MuDan14	501,130,687	150.34
MuDan15	493,013,302	147.9
MuDan2	500,858,291	150.26
MuDan22	491,109,060	147.33
MuDan22	883,333,067	265
MuDan22	3,577,983,239	1073.39
MuDan3	499,220,075	149.77
Total	8,323,929,796	2497.18

-Methods for ChIP-seq and DNA methylation libraries preparation and subsequent analysis are missing. Most method details for Illumina, PacBio, and Hi-C libraries preparation and sequencing are missing.

Response:

Thanks for the comments. This information was added in the method section. Now it reads:

ChIP-seq and DNA methylation analysis

Three replicates of ChIP-Seq were performed to detect genome-wide histone H3 methylation using the anti-histone H3 antibody (tri methyl K27) by Cloud-Seq Biotech (Shanghai, China). Briefly, chromatin immunoprecipitation was performed according to Wamstad et al. (2012)(*Wamstad, J.A., et al., Dynamic and coordinated epigenetic regulation of developmental transitions in the cardiac lineage. Cell. 151(1): p. 206-20.*). The yield of ChIPed DNA was determined via Quant IT fluorescence assay (Life Technologies) and the enrichment efficiencies of ChIP reactions were evaluated by qPCR. Illumina sequencing libraries were generated with NEBNext® Ultra™ DNA Library Prep Kit (New England Biolabs) by following the manufacturer’s manual. The

library quality was checked by Agilent 2100 Bioanalyzer (Agilent), and then subjected to high-throughput 150 bp paired-end sequencing on Illumina HiSeq sequencer according to the manufacturer's recommended protocol.

Seeds of *P. ostii* (two biological replicates) were subjected to reduced-sequencing DNA methylation profiling (short as MethylRAD) to analyze the genome-wide distribution of methylation sites (CpG and CHG) as well as the position of the methylation sites on various gene functional elements. Each sample was tagged with the type II B enzyme FspEI, and the pooled library was sequenced by Illumina 2000 PE (100–150 bp). (Wang S et al. 2015 MethylRAD: a simple and scalable method for genome-wide DNA methylation profiling using methylation-dependent restriction enzymes. *Open Biol.* 5: 150130.).

A total of 61 DNA libraries with a gradient of insert sizes were used for Illumina paired-end sequencing, followed by filtering with SOAPnuke (v1.5.5) (<https://github.com/BGI-flexlab/SOAPnuke>). The 20 kb libraries were constructed for Pacific Biosciences (www.pacb.com) Single-Molecule Real-Time sequencing (SMRT-seq) on both the RSII and Sequel systems, and reads with a length greater than 500 bp were retained for further analysis. Genomic DNA for *in situ* Hi-C libraries was digested with a suitable 4-cutter restriction enzyme (*MboI*), and the ligated fragments were subjected to paired-end sequencing.

-Method for identification of pseudogene is missing.

Response:

Thanks for the comments. Method for identification of pseudogene is added to the manuscript as suggested by the reviewer. Now it reads,

Pseudogenes were identified by the alignment of tree peony protein coding genes to repeat-masked genome using Exonerate (v2.2.0) (<https://github.com/nathanweeks/exonerate>). The predicted hits (with >70% coverage of query proteins) which have frame shift or premature stop codon comparing with the reference proteins were considered as pseudogenes. Finally, we identified total of 330,511 pseudogenes in tree peony genome, and 77,372 were derived from HC genes and 253,139 were derived from LC genes.”

-Method for subcellular localization is missing.

Response:

Thanks for the comments. Method for subcellular localization is added to the manuscript as suggested by the reviewer. Now it reads,

Subcellular localization of FAD3 and functional validation

Fatty acid linolate desaturases (FAD3 and FAD7/8) are key enzymes that catalyze the production of ALA, C18:3 from LA, C18:2, especial FAD3 in peony seed development.

In common, FAD7/8 is located in plastid and FAD3 is located in endoplasmic reticulum (ER). The fused expression vector with green fluorescent protein (GFP) was transferred into *Agrobacterium tumefaciens* using the freeze-thaw method, and simultaneously

infected *Nicotiana benthamiana* leaves.

Based on the Gateway technology, the expression vectors (pDonr207 and Pyes-DEST52 donated by Prof. Qin Zhao, Shanghai Chenshan Botanical Garden) of FAD3 genes (FAD3_1, FAD3_2, FAD3_4) were constructed, and transferred into *Saccharomyces cerevisiae* INVSc1 (Ura- defects, donated by Prof. Zhi-gang Zhou, Shanghai Ocean University). The transformants (pY31 with FAD3_1, pY33 with FAD3_3 and pY34 with FAD3_4) were selected on SC minimal medium that was deficient in uracil (SC-U) and containing 2% glucose as the sole carbon source. The wild type and transgenic yeasts were fed with LA, and 2% galactose was added as an inducer. The fatty acid profiles of yeast were detected by gas chromatography-mass spectrometer (GC-MS) (detailed methods referred to Yu et. Al, Scientific Reports, 2016).

-The author should describe the 448 samples used for GWAS in details.

Response:

Thanks for the comments. More information on 448 samples was added to the manuscript as suggested by the reviewer. Now it reads,

In total, 448 individuals of *P. ostii* from five areas including Luoyang, Tongling, Bozhou, Hezi and Shaoyang of China were introduced in the special Paeoniaceae nursery at Shanghai Chenshan Botanical Garden (31°4'52"N, 121°10'14"E) since 2014. Over four growing seasons from 2016 to 2019, the budded flowers of each of 448 plants were hand-pollinated with the pollen collected from the same *P. ostii* plant and the seeds were

obtained per plant annually.

-L502, “various plant tissues” should provide the name for each tissue.

Response:

Now it reads: Various plant tissues including young leaves and young buds

-L503 “standard protocols from the manufactures”, please add a reference or provide the detail method.

Response:

Now it reads: for DNA extraction using the DNasecure Plant Kit (TIANGEN).

-L514, change “K-meranalysis” to “K-mer analysis”. L555, change “PacBioIso-Seq” to “PacBio Iso-Seq”. In fact, many spaces between word are missing. The authors should check through the manuscript.

Response:

Thanks for pointing this out. These mistakes were checked and corrected.

-In Fig 2f, remove “1e6” on the top.

Response:

Thanks for pointing this out. The figure was corrected as suggested by the reviewer.

-What is colour bar's meaning for Fig.3c and 6b.

Response:

Now it reads: The colour of scale bars stands for relative expression.

-Fig. 5a, how many replicates, and what is the bar represent for?

Response:

After artificial pollination, seed samples of *P. ostii* were collected weekly (7 d) from seven plants over four growing seasons from 2016 to 2019 (in total 13 stages as shown in Fig 5a, six stages' samples labelled with red were selected for RNA-seq, and each stage had two biological replicates). The seed samples were quickly frozen in liquid nitrogen and stored in a freezer at -80°C for later use. The bars represent for standard deviation.

-Fig 6d and Fig S7, how many replicates for the qPCR?

Response:

Three replicates were used for the qPCR.

-Background of Fig. S2, S7, S10 should be white.

Response:

Thanks for pointing this out. The figure was corrected as suggested by the reviewer.

-The font size should be uniform, for example, the font in y axis of Fig 4b and Fig 4e are different.

Response:

Thanks for pointing this out. The figure was corrected as suggested by the reviewer.

-Please provide the accessions for all the raw data for DNA methylation, ChIP-seq, Illumina, PacBio, Hi-C, GWAS etc.

Response:

All the data were uploaded to China National Gene Bank with accession number CNP0003098 (http://db.cngb.org/cnsa/project/CNP0003098_071c0962/reviewlink/).

Reviewer #3 (Remarks to the Author):

Peony has several prominent features, including a giga genome with giga-chromosomes, double flowers that are more attractive and valuable for ornamental crops, and >90% of polyunsaturated fatty acids (PUFA) in seed oil. The authors applied genomic and metabolomic technologies to investigate these intriguing biological questions. They discovered explosive LTR retrotransposon expansion caused massive genome size increase without any recent whole genome duplication after the triplication event shared by all eudicots. The massive amount of histone proteins needed for this giga-genome are supplied by the expansion of gene families encoding the five types of histones, particularly genes encoding H2A.W and H3.1, which promote chromatin condensation, DNA replication, and chromosome assembly. The petaloid-stamens contributing to double flowers are caused by ectopic expression and class A gene AP1 and reduced gene expression of

class C gene AG. GWAS analysis of 448 peony accessions on 35 traits related to fatty acid biosynthesis revealed a cluster of SAD and FAD genes contributing to PUFA biosynthesis. These findings substantially enhanced our understanding of genomic and molecular bases of these biological features. The knowledge and genomic resources gained will benefit plant research community.

Minor changes:

Line 37 - 38: “...but the species still closes to extinction” – ...but the species is close to extinction

Response:

Thanks for pointing this out. The context was corrected as suggested by the reviewer.

Now it reads:

the species is close to extinction in the wild.

Line 85: It would be better to start a new paragraph describing seed oil content.

Response:

Thanks for the comments. As suggested by the reviewer, the context of seed oil content and petaloid-stamen development was separated by starting a new paragraph to distinguish these two parts.

Lines 136 – 137: Only one sentence stating the chloroplast and mitochondrial

genomes were assembled. Although the chloroplast genomes are highly conserved, mitochondrial genomes are variable among flowering plants. I would suggest using a paragraph to describe these two organelle genomes. Because genome evolution is a major part of this manuscript, it is necessary to assess whether organelle genomes evolved at a different pace.

Response:

Thanks for the comments. We performed the evolutionary analysis of *P.ostii* chloroplast genome and a few model plant species. And the result is quite consistent with the genomic analysis except for the position of *V. vinifera*.

Lines 184: “...is tens of times larger...” – It is better to use actual number here.

Response:

Now it reads: is 15 times larger

Line 197: A new paragraph could start from “To explore protein-DNA interactions...”

Response:

We started a new paragraph as suggested by the reviewer.

Line 270: “...and assembled in endoplasmic reticulum...”- ...and assembled in smooth endoplasmic reticulum...

Response:

Thanks for the comment. Now it reads: and assembled in smooth endoplasmic reticulum...

Reviewer #4 (Remarks to the Author):

The paper deals with an interesting phenomenon of genome structure in the peony, which has a relatively large genome but only five chromosomes. I am pleasantly surprised by the set of methodological arsenal used to unravel the nature of giga-chromosomes and have no doubt that the results deserve the attention of the scientific audience. On the other hand, I'm not too happy with some of the evolutionary implications drawn - they are possible, but I don't think you have enough evidence to support them. Meanwhile, some of the claims (especially in the discussion) strike me as outside the scope of the paper, i.e. unnecessary given that

the findings themselves are solid enough. See my specific comments on each section of the text.

Response:

Thanks a lot for the valuable comments. We made serious amendments to eliminate excessive interpretation of the data as suggested by the reviewer, making sure the expression more rigorous in the scientific way. We deleted several sentences which were not evidence-supported, and we are now very careful when we proposed the hypothesis and made the discussion.

Introduction

p. 3, l. 59-62 – The range of genome sizes in Angiosperms should be corrected - the smallest genome size is in *Genlisea aurea* (instead of *G. margaretae* - see Fleischmann et al. 2014 <https://doi.org/10.1093/aob/mcu189>) and the largest is not in *Viscum album* (its genome size in 1C-value, i.e. in the same value as *Genlisea*, is "only" 102.9 pg - Zonneveld 2010, <https://www.hindawi.com/journals/jb/2010/527357/>), but for *Paris japonica* - 1C = 149.2 Gb (Pellicer et al. 2010 - <https://doi.org/10.1111/j.1095-8339.2010.01072.x>). Both values must be in the same form and units (i.e., 1C value in Mb or Gb).

Response:

Thanks for pointing it out. We double checked the references offered by the reviewer and found the genome size from these two plants were indeed the smallest and the

largest. Sorry for the mistake. Now it reads:

Genlisea aurea (Lentibulariaceae) has the smallest known angiosperm genome (63 Mb), while the largest *Paris japonica* (Melanthiaceae) is with 150, 000 Mb.

Results

p. 8-9, l. 169-175 + Table S24 – I recommend unifying the terminology of LTR elements - in the text you use the subtype Gypsy/del, while in table S24 you use the subtype Gypsy/Tekay

Response:

Thanks for the suggestion. The terms were unified as the subtype Gypsy/del and the information in Table S24 was renewed.

p. 9, l. 176-178 – How was the dating of LTR bursts determined? I don't understand the suggested coincidence with volcanic eruptions (see also comments to Discussion).

Response:

Sorry for the confusing. LTRretriever was used to tell the distance K between the 5'-LTR and 3'-LTR of each complete LTR retrotransposon, and the LTR insertion time (T) was counted based on formula $T = k/2r$. As there is no solid evidence on the relationship with volcanic eruptions, it is not proper to write these views here as pointed out by the reviewer, therefore these words were deleted. Now it reads:

The LTR burst in *P. ostii* occurred 1-2 Mya, with a peak at 1.4 Mya (Fig. 2b/2d).

p. 9, l. 181-183 – I don't understand the meaning - the enzymes for synthesis of del were 8.25 times more than what? I assume “than for synthesis of other LTRs”, but the meaning is different.

Response:

Thanks for the comment. Sorry for the confusing. Now it reads:

Further analysis indicated that the prevalence of five enzymes that are critical for del synthesis was 8.25 times higher than for synthesis of other LTRs

p. 13, l. 283-285 – Interpretation of results belongs to Discussion section

Response:

Thanks for the comment. Now it was moved to Discussion.

p. 14, l. 294-304 – I don't understand the placement of this paragraph in the results - it should be in the introduction or discussion. The results section should be dedicated to the results only. In addition, please omit the adjective "beautiful" if you are talking about petals. The whole Petaloid-stamen results section is actually written differently than the other sections, combining results with discussion, which is peculiar in the overall context of the other sections.

Response:

Thanks for the comments. It is not proper to leave these words here in results as suggested by the reviewer. The sentences about ABCE model were transferred to

discussion. We checked the grammars in petaloid-stamen results section and made a few optimizations in writing this part. Thanks again.

discrepancy between Table S23 and S24 – I don't understand the calculation of the proportion of the genome made up of LTR elements. In Table S23 you state that LTR retrotransposons in *P. ostii* make up ~43% of its genome. At the same time, in Table S24, you attribute only ~12% to the proportion of LTRs in the genome.

Response:

Thanks for pointing this out. Table S24 describes the full-length LTRs which are parts of the total LTRs in Table S23. As the genome includes fragmented LTRs in total, but when comparing with other plant species, the full-length LTRs are more representative. Sorry we did not specify this in the manuscript. The title of Table S24 is now changed to: Distribution of full-length LTR subtypes in land plants. The main text now reads: reaching 70.58% of full-length LTRs. And fig2c legend: Subgroups of full-length LTRs in selected Gymnospermae, Dicotyledon and Monocotyledon species.

Discussion

p. 17, l. 339 – How does your statement "*P. ostii* has the largest chromosome size known in plants sequenced to date" relate to the cited Hidalgo et al. 2017 paper? As far as I know, the paper deals with huge genomes in plants (more than 100 Gb), which is far beyond the relatively common genome size of *Paeonia* (~12 Gb). Moreover, the paper does not discuss chromosome size in sequenced plants at all.

Response:

Thanks for the comments. The Hidalgo et al. 2017 paper talked about giant genomes in flowering plants and ferns, but indeed this paper did not directly specify the information we mentioned here. It is not a proper reference here as suggested by the reviewer, therefore it was deleted. Now it reads:

To our knowledge, *P. ostii* has the largest chromosome size known in plants sequenced to date.

p. 17, l. 344-348 – I found no evidence in the paper that provides a basis for such a strong statement. Could you explain, not only in the discussion but also in the M&M and Results sections, on what basis you date the expansion of LTR to the recent 1-2 My?

Response:

Thanks for the comments. As mentioned above, LTRretriever was used to tell the distance K between the 5'-LTR and 3'-LTR of each complete LTR retrotransposon, and the LTR insertion time (T) was counted based on formula $T = k/2r$. Still we deleted the word: 'in a very short period (the last one to two million years)' and 'recently' as the reviewer thought it is not rigorous, now it reads:

We found that the giga-chromosomes of the peony genome seem to have been driven by large-scale LTR expansion in the intergenic regions, indicating that the giga-chromosomes and giga-genome were formed with adaptive evolution.

p. 18, l. 365-368 – You repeatedly talk about volcanic events in the last two million years and their relation to adaptive evolution in Paeonia, but you never specify what you mean by "volcanic events" and which ones are essential for the evolution of giga-chromosomes. Are you sure you know what you're talking about? Do you really mean volcanic, i.e. volcanic eruptions?

Response:

Thanks for the comments. As mentioned above, there is no solid evidence on the relationship with volcanic eruptions, it is not proper to write these views here as pointed out by the reviewer. The whole sentence was deleted to reduce confusion.

p. 18, l. 368-375 – I'm not too familiar with the way you're conducting the discussion. Have you analyzed the seed set in relation to giga-chromosomes or genome size? I don't think so, so why are you going down such a speculative path that is completely out of line with the very interesting findings you have made? Why you repeatedly use the term giga-genome? The size of the genome of Paeonia ostii is not gigantic, its genome is only one of the larger ones among plants.

Response:

Thanks for the comments. As pointed out by the reviewer, we did not perform the analysis on seed set in relation to giga-genome size. The reason we write these words here was to propose an idea on the relation of genome size and plant reproduction, but it was supported by the data from our paper. As suggested by the reviewer, it is not proper to make such strong paraphrase without evidence, we therefore deleted the

words of line 368-375. The authors used the term giga-genome for the whole paper as there was rare plants which were sequenced with large genomes (above 10G), but we believe there would be more and more plants with large genomes sequenced at giga-size level. Although there were challenges like high duplication and high heterozygosity, hopefully our cutting-edge work would encourage more researcher to work in the area of large genomes like giga-genome.

Methods

Take care to write plant names in strict binomial form - *Vitis vinifera* instead of *Vitisvinifera*, *Malus domestica* instead of *Malusdomestica*, etc.

Response:

Thanks for the comments. The plant names were corrected as suggested by the reviewer.

Figures

Fig. 2 – As all images should be easy to understand (self-explanatory), abbreviated plant names should be explained in the caption.

Response:

Thanks for the comments. The plant names were corrected as suggested by the reviewer.

Fig. 5c – The meaning of the red asterisk and the triangle in phylogeny is not clear.

Response:

Thanks for the comments. The figure legend was corrected as suggested by the reviewer.

Tables

In several cases (Table S19, S24) you used only abbreviated plant names (only the first letter of the genus name), which is difficult to understand.

Response:

Thanks for the comments. The plant names in Table S19 and S24 were corrected as suggested by the reviewer.

Reviewers' Comments:

Reviewer #1:

Remarks to the Author:

I am happy to see that the fatty acids are correctly identified and more details for couple of methods are provided. I have no further suggestion.

Reviewer #2:

Remarks to the Author:

I appreciate that most of my concerns have been addressed by the authors, and the current form of manuscript was largely improved. Considering that no statistics support for the relationship between number of histone variants and chromosome size, I suggest the authors further tone down the claims to avoid delivering unsupported message that expansion of histone number is maintaining or critical factor for the large chromosome in *Paeonia ostia*.

Reviewer #3:

Remarks to the Author:

The authors addressed my concerns. No more comment.

Reviewer #4:

Remarks to the Author:

I am convinced that the authors have dealt very well with all the reviewers' comments and have incorporated them properly into the new version of the manuscript. In quite a few cases they have argued well why they did not do so and why they chose a different approach than that recommended by the reviewers. For the reasons described above, I have no need to comment further on the manuscript and highly recommend it for publication.

REVIEWERS' COMMENTS

Thanks for the comments. The corresponding responses were listed below point-by-point.

Reviewer #1 (Remarks to the Author):

I am happy to see that the fatty acids are correctly identified and more details for couple of methods are provided. I have no further suggestion.

Response:

Thank you very much for the valuable comments, these suggestions did help us a lot to improve the manuscript. It is much appreciated.

Reviewer #2 (Remarks to the Author):

I appreciate that most of my concerns have been addressed by the authors, and the current form of manuscript was largely improved. Considering that no statistics support for the relationship between number of histone variants and chromosome size, I suggest the authors further tone down the claims to avoid delivering unsupported message that expansion of histone number is maintaining or critical factor for the large chromosome in Paeonia ostia.

Response:

Thanks for the comments. As pointed out by the reviewer, there is no solid evidence on

the relationship between number of histone variants and chromosome size, it is not proper to write these views here. The whole sentence and the related graph were deleted to reduce confusion. The authors would try to initiate more verifications in the upcoming future.

Reviewer #3 (Remarks to the Author):

The authors addressed my concerns. No more comment.

Response:

Thank you very much for reviewing the manuscript.

Reviewer #4 (Remarks to the Author):

I am convinced that the authors have dealt very well with all the reviewers' comments and have incorporated them properly into the new version of the manuscript. In quite a few cases they have argued well why they did not do so and why they chose a different approach than that recommended by the reviewers. For the reasons described above, I have no need to comment further on the manuscript and highly recommend it for publication.

Response:

The authors appreciate the valuable comments from the reviewer. All authors did work hard to revise the manuscript, and it is worth to do so.